# From Takeoff to Touchdown: A Decade's Review of Carbon Emissions from Civil Aviation in China's Expanding Megacities

**Ying She** [1,2]**, Yangu Deng** [1,]*** and Meiling Chen** [1]

1    School of Economics and Management, Nanchang Hangkong University, Nanchang 330061, China;
sheying@nchu.edu.cn (Y.S.); 2209120100022@nchu.edu.cn (M.C.)
2    Research Center for International Educational Comparative Studies, Nanchang Hangkong University,
Nanchang 330061, China
*    Correspondence: deng_yangu@nchu.edu.cn

**Abstract:** The rapid growth of urbanization in China has led to a substantial escalation in the demand for civil aviation services, consequently propelling China to the third-largest contributor of carbon emissions within the aviation sector. Using the 2012–2021 data on takeoffs and landings of civil aviation aircraft in China, the aircraft engine emission factor database of the Base of Aircraft Data (BADA) from EUROCONTROL, this paper investigates the spatial-temporal distribution characteristics of atmospheric pollutants, primarily carbon emissions from Chinese civil aviation aircraft in 19 megacities. The results indicate that (1) China's aviation $CO_2$ emissions equivalent between 2012 and 2022 has been on an upward trajectory, peaking at 186.53 MT in 2019 with an average annual growth of 12.52%. The trend, albeit momentarily interrupted by the COVID-19 pandemic, appears to persist. (2) $CO_2$ constitutes the highest proportion of aircraft emissions at 83.87%, with Cruise Climb Descent (CCD) cycle emissions accounting for 96.24%. $CO_2$ and $NO_X$, with the highest increase rates in the CCD and Landing and Takeoff (LTO) phases, respectively, are identified as the chief culprits in aviation-related greenhouse effects. (3) There is a marked spatial imbalance, with 19 megacities contributing 62.08% of total $CO_2$ emissions, compared to the 207 least-emitting cities contributing just 9.29%. (4) The pattern of city carbon emissions is changing, with rapid growth rates in the western cities of Xinjiang, Tibet, Shaanxi, and Guizhou, and varied growth rates among megacities. The implications of this study emphasize the urgency for advancements in aviation fuel technology, rigorous management of CCD phase pollutants, strategic carbon emission controls in populous cities, fostering green aviation initiatives in western regions, diverse carbon mitigation tactics, and strengthening the precision and surveillance of aviation carbon accounting systems. Collectively, this study paints a grand picture of the complexities and challenges associated with China's urban sprawl and aviation carbon emissions.

**Keywords:** carbon emission; civil aviation; LTO phase; CCD phase; megacity; China

## 1. Introduction

The aviation sector, an integral pillar of national economic advancement, is confronting an escalating imperative for carbon reduction. According to data from the International Energy Agency (IEA), in the three decades preceding the outbreak of the COVID-19 pandemic in 2019, carbon dioxide emissions originating from the global transportation sector exceeded those engendered by industrial activities. This made transportation the second-largest source of carbon emissions worldwide, just behind the electricity and heat production sector [1]. Notably, the growth rate of carbon emissions within the aviation sector has outstripped that of roadways, railways, and maritime transport, which has been identified as an important source of carbon dioxide ($CO_2$) emissions, releasing approximately 900 million metric tons in 2019 [2,3]. This figure is expected to nearly double by the year 2035 according to the continued growth of air transport [4]. By 2050, the aviation sector



might contribute to 25% of the global carbon budget [5]. Due to economic development and increased urbanization, China's aviation industry has rapidly expanded, making it the world's third-largest emitter [6]. From 2013 to 2019, China's carbon emissions increased by 66% [6,7]. The Chinese government initiated carbon reduction policies for the aviation sector relatively late. In 2022, the Civil Aviation Administration of China released the "14th Five-Year Plan for Green Development of Civil Aviation", outlining the goal for airport carbon dioxide emissions to reach a peak and stabilize by 2035 [8]. In 2023, the Ministry of Industry and Information Technology, the Ministry of Science and Technology, the Ministry of Finance, and the Civil Aviation Administration of China issued the "Green Aviation Manufacturing Industry Development Outline (2023–2035)" [9]. This document emphasizes the primary focus on small electric aircraft and actively explores technological pathways for hydrogen energy and Liquefied Natural Gas (LNG). However, it does not explicitly set targets for aviation carbon reduction. This study aims to accurately measure carbon emissions by the aviation sector in China, the largest developing country and the most potential market for civil aviation. Such measurement is crucial for analyzing future emission scenarios in the aviation industry, exploring optimal emission reduction pathways, implementing refined emission reduction measures, and formulating carbon peaking strategies.

Despite accounting for just 2.9% of total emissions, the carbon reduction in the aviation sector faces significant challenges due to three primary factors [10]. Firstly, unlike other transportation modes like road or rail, the implementation of new energy technologies for carbon reduction is inherently difficult for civil aircraft. Secondly, unlike emissions from ground-level sources, aircraft exhaust is discharged directly into the upper troposphere and stratosphere, amplifying its greenhouse effect. Lastly, the concentration of populations in major urban areas in developing countries has resulted in a surge in air travel demand, outpacing the rate at which advancements in aviation technology can mitigate carbon emissions [11].

China is a vast country with an expansive E-W development of approximately 5200 km and an N-S extension of approximately 5500 km. Meanwhile, China is also the most populous nation globally. Especially, with the rapid pace of urbanization, China's megacities are experiencing accelerated expansion. According to the Seventh Population Census data, China boasts 7 cities with urban populations exceeding 10 million (Shanghai, Beijing, Shenzhen, Chongqing, Guangzhou, Chengdu, and Tianjin), and 14 cities featuring urban populations between 5 and 10 million (Wuhan, Dongguan, Xi'an, Hangzhou, Foshan, Nanjing, Shenyang, Qingdao, Jinan, Changsha, Harbin, Zhengzhou, Kunming, and Dalian). The aggregate population of these sizable urban hubs constitutes 20.7% of the nation's total populace. Functioning as regional epicenters for economic, political, cultural, and social matters, these vast metropolises also play a critical role as central hubs in civil aviation [12]. The combination of expansive territorial coverage and substantial urbanization signifies the increasing demand for civil aviation in China.

As urban centers expand and more cities emerge as key hubs, air connectivity between these cities has grown, driven by both business and leisure travel demands. New airports and routes have been introduced to accommodate this surge in demand. As a direct consequence, flight frequencies have increased by approximately 10% annually over the past decade, leading to elevated levels of aviation carbon emissions. While new airports and routes accommodate this growth, their construction and energy-intensive processes also contribute to emissions. According to the "14th Five-Year Plan for Civil Aviation Development", the Compound Annual Growth Rate (CAGR) of China's civil aviation passenger traffic from 2019 to 2025 is projected to be 5.9%. By 2025, the passenger traffic is expected to reach 930 million [13]. Currently, China's civil aviation industry is still in a growth stage, and with further urbanization, there is significant potential for increasing the per capita air travel frequency. Thus, an interesting question has been posed: What are the spatiotemporal trends in carbon emissions in China's civil aviation sector? Specifically, as

China's rapidly developing mega-cities have a substantial demand for civil aviation, how fast are carbon emissions from aviation growing in these metropolises?

The significant growth of carbon emissions from the aviation sector has led to a wide discussion in previous literature [6,14–16]. International organizations such as the International Civil Aviation Organization (ICAO), the European Environment Agency (EEA), and the U.S. Environmental Protection Agency (EPA) have proposed calculation methodologies for civil aviation carbon emissions, which scholars have utilized in the computation of carbon emissions at specific airports, at the country level, and globally [17,18]. The advanced approach of the ICAO, considering the pollutant emissions from different aircraft and engine types at various flight stages, is the most widely used, deemed the "bottom-up" method. This methodology is similar to that of the U.S. EPA [19–21]. However, the ICAO method focuses only on the emissions during the Landing and Takeoff (LTO) phase. According to the International Air Transport Association (IATA) data, LTO phases account for approximately 10% of all emissions during flight [22]. Another method widely used is the European Monitoring and Evaluation Program (EMEP) by the European Environment Agency (EEA). This method, considered a "top-down" approach, estimates $CO_2$ emissions based on total fuel consumption during flight, without the need to consider differences between different flight engines. Its advantage lies in its convenience, as it does not require detailed information about flight duration, aircraft type, or engine. However, its precision is somewhat lacking. Additionally, ICAO's EEDB, Aircraft Engine Emissions Databank, and EUROCONTROL's BADA database provide the aircraft carbon emission coefficients for both the Landing and Take Off (LTO) phases and Cruise Climb Descent (CCD) phases. Based on these methods and data, numerous studies have carried out precise calculations of civil aviation carbon emissions for specific countries, regions, or globally.

Due to data limitations, some research focuses only on aviation carbon emissions during the LTO phase. For instance, using the EMEP/CORINAIR methodology, a study on carbon emissions from Greece's domestic and international flights from 1980 to 2005 found that aviation carbon emissions increased with air traffic growth [23]. However, the rate of increase varies across airports, and changes in fleet composition and each airport's contribution to total air traffic can impact the increase in each air pollutant [24]. Makridis and Lazaridis (2019) used ICAO's method and daily data from air traffic to calculate the aircraft pollution emissions of Chania Airport (Crete) LTO cycles in 2016. They discovered that $NO_2$ exceeded the standard in the busy summer tourist season, but $CO$, $SO_2$, and PM2.5 did not [23]. Most research blends the "bottom-up" approach, suggested by the ICAO to calculate Landing and Takeoff (LTO) emissions, and the "top-down" method proposed by the EMEP for total aviation carbon emissions to ensure complete and precise outcomes. Puliafito (2023) used the EMEP method to calculate Argentina's civil aviation LTO and CCD cycle carbon emissions and found that monthly $CO_2$ emissions ranged from 6700 t to 179,000 t from 2001 to 2019 [18]. As a result of fleet renewal, the energy efficiency index improved from 308 g $CO_2$eq/Revenue Passenger Kilometer (RPK) to 107 g $CO_2$eq/RPK. In a distinctive analysis, Eskenazi et al. (2022) calculated emissions for LTO and CCD stages using flight data from the United States' third quarter of 2021 [25]. The study discovered substantial variances in emission levels on specific routes owing to the choice of aircraft and engine, as well as significant differences in emission rates among different airlines. Upon examining aviation carbon emissions, several studies have further explored carbon mitigation in aviation from various angles. One such exploration focuses on the benefits of increasing direct flights while reducing stopovers to cut carbon emissions. Using data from the ICAO, Debbage (2019) studied the carbon emissions of direct and connecting flights between ten of the largest cities in the northeastern United States and thirteen travel destinations in the U.S. Sun Belt and west. The results showed that, on average, direct flights reduce carbon emissions by approximately 100 kg per passenger compared to connecting flights [26]. Another stream of literature focuses on the Carbon Offsetting and Reduction Scheme for International Aviation (CORSIA). In 2016, the International Civil Aviation Organization (ICAO) introduced the CORSIA. Notably, the top ten carbon-

emitting countries, including the United States, China, Japan, Brazil, India, Indonesia, Russia, Australia, Canada, and the European Union, voluntarily joined this scheme. If these countries also included their domestic flights in the CORSIA plan, it could lead to a long-term reduction of an additional 50% in carbon dioxide emissions [11]. Analyzing the ICAO's carbon offset program from environmental, economic, and social perspectives reveals that sustainable aviation fuels (SAFs) have great potential to reduce climate impacts. However, they are still in the early stages of widespread use in the aviation sector, with challenges such as expanding farmland [27].

Although China is the third-largest civil aviation carbon emitter globally, research focusing on China is still relatively scarce. In the recent research literature, there has been a gap in the method used to calculate aviation carbon emissions. Traditionally, studies estimating aviation carbon emissions have uniformly adopted the International Civil Aviation Organization (ICAO) methodology for the Landing and Takeoff (LTO) cycle, encompassing stages such as take-off, climb, descent, and taxiing. However, for the Cruise, Climb, and Descent (CCD) cycle, methodologies vary, with the most prevalent approach being rooted in fuel consumption metrics. Using the Fuel Percentage Method for the CCD phase, Liu et al. (2019) made a detailed emission inventory of various airborne pollutants from China's civil aviation from 1980 to 2015. Their work shows a rapid increase in annual emissions, in line with China's growing economy and population. By 2015, emissions were estimated to be 4.77 KT for hydrocarbons (HC), 59.63 KT for carbon monoxide (CO), 304.77 KT for nitrogen oxides (NOx), and a significant 59,961 KT for carbon dioxide ($CO_2$). Importantly, 81% of $CO_2$ came from the CCD cycle, while 76% of HC and 71% of CO were mainly from the LTO cycle [28]. Applying the same method, Li et al. (2022) created an urban aviation carbon emissions network. They found the Shanghai–Beijing, Beijing–Shenzhen, and Beijing–Guangzhou routes had the highest carbon outputs [29]. Zhou et al. (2016) used simulations to suggest that without major tech advances, China's civil aviation might not reach carbon neutrality by 2030 [20]. Also, routes with notable increases in carbon emissions included Korla–Urumqi, Dalian–Qingdao, Kunming–Lijiang, Shanghai–Wenzhou, and Xishuangbanna–Lijiang [30]. By using the Modified Fuel Percentage Method (MFPM) to study the CCD cycle's carbon emissions, the difference between the results and official data was around 6.45%. Also, the "13th Five-Year Plan" for energy saving in civil aviation did not seem to help reduce China's domestic aviation carbon emissions [31].

Precise estimation of specific airport carbon emissions necessitates more refined methodologies. An improved method, considering the mixing layer height calculated based on the Aircraft Meteorological Data Relay (AMDAR), instead of using the height (915 m) recommended by the ICAO, was used to calculate the pollution emissions during the LTO phase at Beijing Capital International Airport. The study found that NOx was mainly concentrated in the take-off and climb phases, while CO and HC were concentrated in the taxi phase, accounting for 91.6% and 92.2% of the total emissions, respectively [32]. Based on China's 2010 flight schedule and the Boeing Fuel Flow Method 2, the fuel consumption and emissions of HC, CO, NOx, $CO_2$, and $SO_2$ from China's domestic flights (excluding Taiwan province) in 2010 were estimated. The empirical evidence indicates that, in 2010, domestic aviation in China witnessed a fuel consumption of 12.12 million metric tons. Among the aviation entities, China Southern Airlines bore a significant environmental footprint, accounting for a substantial 27–28% of the total pollutant emissions [14]. The above studies have adopted diverse methodologies to estimate the emissions of aviation pollutants from a specific airport in China, as well as on a nationwide scale. These studies typically rely on cross-sectional data or are conducted over relatively brief periods and overlook the significant aviation pollution problems posed by megacities.

Using the established methods for calculating civil aviation carbon emissions, researchers have further explored the factors influencing these emissions, possible reduction policies, scenarios for reaching carbon neutrality, and the relationship between carbon emissions and the growth of civil aviation. Liu and Zhang (2023) used an index decomposition analysis method to study the relationship between carbon emissions and the growth of

demand in China's civil aviation. Their results showed that even though the increase in aviation demand led to more energy use, the total carbon emissions from aviation were growing, but the rate of these emissions was decreasing [28]. Changes in energy use and transportation methods were identified as the main reasons for this trend. Building on this, Yang et al. (2023) used the Delphi method, setting scenarios that considered uncertainties like aviation growth and emission reduction policies. They found that for the global aviation industry to reach a net-zero carbon emission goal, China would need to cut its emissions by roughly 82–91% based on the best-case scenarios. This points to the significant challenge China's civil aviation industry faces in reducing emissions. By 2050, using sustainable aviation fuels stands out as the best way to reduce these emissions [33].

In this context, this article tries to make marginal contribution in the following three ways: (1) using the comprehensive compiled panel data from three sources: the daily flight data on takeoffs and landings of civil aviation aircraft in China, the aircraft engine emission factor database of the Base of Aircraft Data (BADA) from EUROCONTROL, this paper evaluates carbon emissions of civil aviation sector in the largest developing country from 2012–2021; (2) in contrast to prior research that solely incorporates aircraft types and engines in a bottom-up carbon emission estimation during the LTO phase, while adopting a fuel consumption extrapolation in the CCD phase, this study consistently employs a bottom-up approach for both LTO and CCD phases, and by doing so, it evaluates carbon emissions for each individual flight, thereby ensuring a more precise outcome; (3) considering China's expanding urbanization, calculating carbon emissions of civil aviation in China's 19 megacities with population more than 5 million.

The remainder of this paper is structured as follows. Section 2 briefly presents the research area. Section 3 lays out the data collection and the details of the methodology, including the consideration of different flight phases. Section 4 analyzes the calculation results. Section 5 concludes the study and provides policy implications.

## 2. Research Area

### 2.1. China's Megacities

Our study concentrates on aviation carbon emissions in China's megacities. According to the United Nations, cities with a population of over 1 million are typically classified as megacities. In 2022, there were 7 urban agglomerations, namely Shanghai, Beijing, Shenzhen, Chongqing, Guangzhou, Chengdu, and Tianjin, boasting a population exceeding 10 million, thereby classifying them as super large cities. Meanwhile, 14 urban territories, including Wuhan, Dongguan, Xi'an, Hangzhou, Foshan, Nanjing, Shenyang, Qingdao, Jinan, Changsha, Harbin, Zhengzhou, Kunming, and Dalian, with their populations ranging from 5 million to 10 million, fall into the category of large cities. Super large and large cities are collectively referred to as "megacities" in this study.

China's megacities serve as the nation's epicenters for manufacturing and technological innovation. Notably, Beijing, Shenzhen, and Shanghai, with per capita GDPs surpassing 18,000 RMB (equivalent to $2500 USD), are critical contributors to the nation's burgeoning economic narrative. Representing the largest demographic swath, characterized by unparalleled population influx, these regions epitomize the zenith of China's prosperity and rapid development. Their prominent role in domestic and international economic dialogues has also positioned them as pivotal demand centers for civil aviation. The megacities experience a pronounced demand for civil aviation, resulting in approximately 70% of the nation's flight takeoffs and landings.

However, Dongguan lacks an airport due to its proximity to Guangzhou. Foshan's airport, though operational, has a modest scale, boasting a mere 18 routes. Between 2009 and 2018, it functioned as a civil airport but otherwise remained dormant for maintenance or served military purposes. Consequently, for the purposes of this study, major metropolitan areas are defined as the 19 cities excluding Foshan and Dongguan.

## 2.2. Historical Evolution of Civil Aviation in China's Megacities

According to the flight schedule data from the Civil Aviation Administration of China (CAAC) [34], from 2013 to 2021, there is a discernible upward trend in the total number of national civil aviation flights, climbing from 2.95 million to 5.2 million instances, as depicted in Figure 1. Metropolises display relatively stable growth, registering an increase from 0.57 million to 0.76 million flights. Conversely, major urban centers see a sharper rise, advancing from 1.70 million to 2.95 million. Together, these two categories of megacities consistently represent approximately 70% of the total flight count. It is noteworthy that Figure 1 illustrates the planned flight frequency as outlined by the Civil Aviation Administration of China (CAAC). In practice, China's civil aviation flight operations decreased by 25.30% and 28.49% in 2020 and 2021, respectively, compared to the 2019 figures [32]. Subsequent carbon emission computations in this article were quantified upon conversion of the planned schedule.

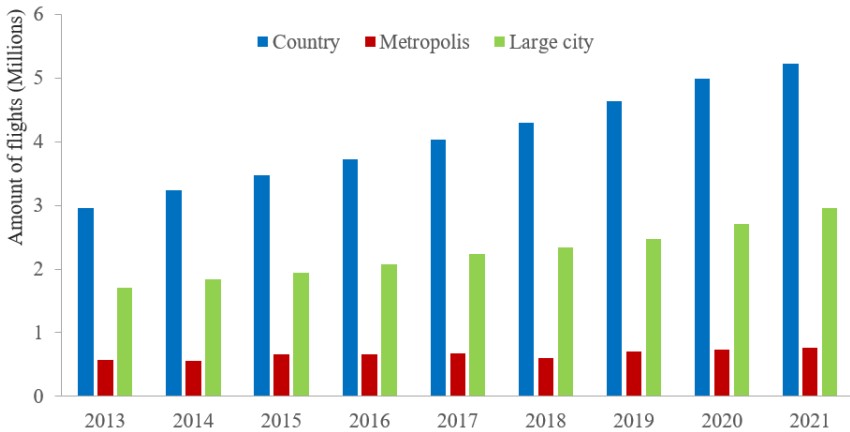

**Figure 1.** Flight Frequency in China's Megacities (2013–2021).

## 2.3. Spatial Characteristics of Civil Aviation in China's Megacities

According to the flight schedule data from the Civil Aviation Administration of China (CAAC), among the 19 mega-cities, there is a general positive correlation between city population and the number of flight movements: cities with larger populations tend to have more flights. Figure 2 categorizes these 19 mega-cities based on population-flight movements into 4 distinct tiers.

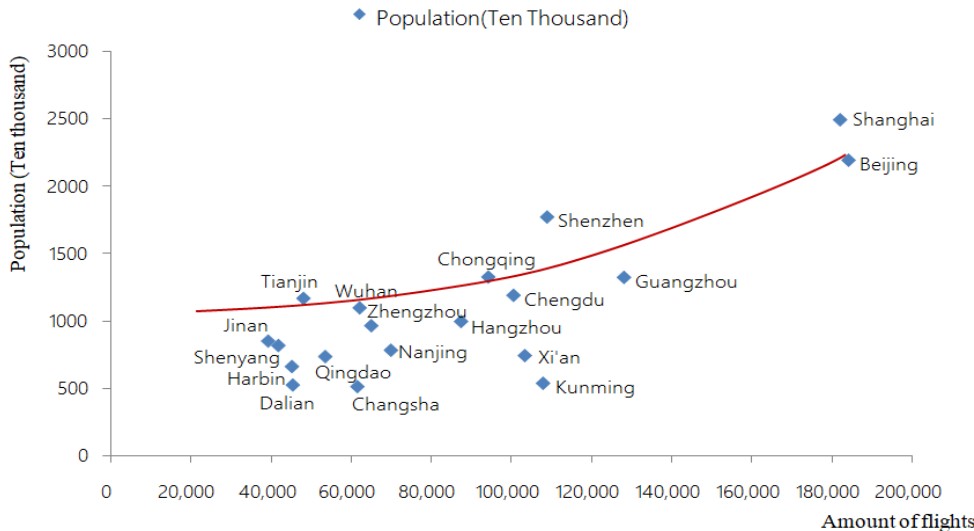

**Figure 2.** The relationship between population and flight frequency.

The foremost tier is represented by Beijing and Shanghai. Serving as the economic, political, and cultural centers of China, these cities house the nation's largest and busiest airports, namely Beijing Capital International Airport, Beijing Daxing International Airport, Shanghai Hongqiao International Airport, and Shanghai Pudong International Airport. Their flight movements consistently occupy the top two positions, registering 183,986 and 181,856 flights, respectively.

The second tier includes Guangzhou and Shenzhen, both of which are pivotal first-tier cities and crucial gateways for international openness. Between 2012 and 2021, their aggregate flight movements tallied at 128,003 and 108,849, respectively. Kunming, Xi'an, Chengdu, and Chongqing, paramount provincial capitals in the western part of China and recognized tourist hubs, fall within the third tier. Their flight movements range between 94,192 and 107,830.

The final tier encompasses Hangzhou, Nanjing, Zhengzhou, Wuhan, Changsha, Qingdao, Tianjin, Dalian, Harbin, Shenyang, and Jinan. These cities, which are vital provincial capitals and economic powerhouses in the eastern and central parts of China, boast flight movements that, at a minimum, reach 39,316.

### *2.4. Airplane Model Structure*

Over the same period, the diversity of aircraft types at the airports of 21 major Chinese cities contracts from 18 to 36 distinct models. Although the repertoire of aircraft variants is vast, the utilization frequency of each model exhibits pronounced disparities. An analysis of the air traffic composition, based on data compiled by the Civil Aviation Administration of China (CAAC), reveals distinct patterns. Among the extensive range of aircraft types (amounting to 49 in total), a mere 10 variants execute the lion's share of LTO cycles, accounting for 94% of the total air traffic. The residual 6% pertains to aircraft with a modest footprint, each recording fewer than 40,565 flights. The dominance of Boeing and Airbus is evident, jointly amassing over 92.91% of the traffic—with respective shares of 47.68% and 45.23%. Specifically, the A320 and B738 models are predominant, contributing 35.36% and 27.10%, respectively.

## 3. Data Description and Methodology

The current section examines the quantification of $CO_2$ emissions from civil aviation. A comprehensive civil aviation flight comprises the landing and take-off cycle and cruise phase [33]. Each flight phase possesses unique characteristics, resulting in varying fuel consumption and carbon emissions at different stages. We aim to quantify the carbon dioxide ($CO_2$), carbon monoxide (CO), hydrocarbons (HC), and nitrogen oxides ($NO_x$) emissions from domestic passenger flights in China's 19 megacities from 2012 to 2021. To obtain the overall impact of aviation pollutants on the greenhouse effect, we also convert the aforementioned pollutants into carbon dioxide equivalents.

### *3.1. Data Description*

This paper aims to calculate the carbon emission of civil aviation aircraft in China's megacities using micro-level data from 4 major sources.

(1)    Schedule database

The schedule data utilized for estimating emissions are derived from the Civil Aviation Administration of China (CAAC), which provides information on aircraft types, flight schedules, departure airports, arrival airports, departure times, and arrival times. The CAAC provides scheduling information for both domestic and international flights spanning 32 provinces, autonomous regions, and municipalities in China. However, our research specifically narrows its focus to domestic flights, excluding those to and from international destinations, Hong Kong, Macau, and Taiwan. The period under examination extends from 2012 to 2021. Table 1 presents the descriptive statistics for the main variables provided by the CAAC.

**Table 1.** Summarize the statistics of the CAAC variables.

| Year | No. of Airport | No. of City | Flight Frequency | No. of Aircraft Type |
|------|----------------|-------------|------------------|----------------------|
| 2012 | 183 | 171 | 2,851,784 | 33 |
| 2013 | 193 | 184 | 3,126,682 | 37 |
| 2014 | 202 | 207 | 3,345,290 | 33 |
| 2015 | 208 | 205 | 3,571,880 | 49 |
| 2016 | 219 | 226 | 3,870,282 | 38 |
| 2017 | 229 | 233 | 4,187,014 | 37 |
| 2018 | 235 | 241 | 4,445,948 | 37 |
| 2019 | 239 | 233 | 4,751,682 | 31 |
| 2020 | 242 | 237 | 5,549,934 | 26 |
| 2021 | 248 | 237 | 5,821,764 | 23 |

(2) Engine type

The selection of aircraft–engine combinations is sourced from the official websites of aircraft manufacturers. It is noteworthy that a specific type of aircraft, under the same airline, could be equipped with different types of engines. However, access to detailed aircraft-engine combination data remains challenging. To streamline the analysis, we opted for the most typical combinations based on the synthesis of information from both aircraft manufacturers and the International Civil Aviation Organization's Emissions Database [34]. Table 2 presents the aircraft–engine combinations employed for computation in this study.

**Table 2.** Typical aircraft/engine combinations used for calculation.

| Aircraft Type | Engine Type | Frequency (Times) |
|---------------|-------------|-------------------|
| B738 | CFM56-7B24 | 14,481,012 |
| A320 | V2500-A1 | 11,101,090 |
| A319 | CFM56-5B8/P | 4,255,836 |
| B737 | CFM56-3C-1 | 3,193,190 |
| A321 | CFM56-5B3/3 | 3,038,984 |
| E90 | CF34-10E5 | 1,473,030 |
| CR9 | CF34-8C5 | 694,044 |
| A330 | CF6-80E1A1 | 459,862 |
| B733 | CFM56-3-B1 | 397,020 |
| A333 | Trent 768 | 387,478 |
| MA60 | PW1127G-JM | 293,332 |
| A332 | Trent 768 | 223,522 |
| B787 | Trent 1000-A | 203,658 |
| ERJ | AE3007A | 171,938 |
| CR2 | CF34-3B | 155,012 |
| ARJ | CF34-10A16 | 114,764 |
| B752 | RB211-535E4 | 104,390 |
| B777 | GE90-115B | 74,880 |
| B788 | GEnx-1B54 | 71,136 |
| A350 | Trent XWB-97 | 61,828 |

(3) Aircraft Engine Emission Database

The third major data source is the Aircraft Engine Emission Database (EEDB) of the International Civil Aviation Organization (EEDB, Aircraft Engine Emission, ICAO), which includes pollutant emissions indices for each type of aircraft engine during the Landing and Takeoff (LTO) cycle. The emission indices are measured under the International Standard Atmosphere (ISA) condition at sea level. The pollutants of CO, $NO_X$, and HC can be directly obtained from EEDB, while the $CO_2$ considered in this paper is that for every kilogram of jet fuel consumed, 3.155 kg is emitted [12].

(4)　Engine emission indices

The fourth major data source is the BADA database (BADA, Base of Aircraft Data) from EUROCONTROL, which provides pollutant emissions (CO, $NO_X$, HC) and $CO_2$ for different aircraft models during the Climb, Cruise, and Descent (CCD) cycle. The BADA database is updated every 3 years, and the latest version is 2023, which has been chosen in this study. EEDB and BADA have widely been used in previous literature [14,23,35].

*3.2. Methodology*

The emissions from a complete flight vary at stages, as the engine operates at different thrusts in each flight stage. To precisely quantify these pollutant emissions throughout a flight, the ICAO has defined and categorized two distinct flight stages: Landing and Take-off (LTO) and Cruise, Climb, and Descent (CCD). The initial cycle, LTO, encapsulates all flight activities occurring within an altitude of 3000 ft (equivalent to 915 m) from the ground. In contrast, the CCD represents all activities that transpire beyond this 3000 ft altitude. For the purposes of this study, carbon emissions from civil aviation are calculated for the aforementioned stages and subsequently aggregated.

3.2.1. Emissions during the Landing and Take-Off (LTO) Flight Cycle

The ICAO gives three distinct methodologies for quantifying emissions during the Landing and Take-Off (LTO) phase: the Simple Approach, the Advanced Approach, and the Sophisticated Approach. While the Simple Approach offers the virtue of simplicity, demanding minimal data, its estimations are notably less precise. Conversely, the Sophisticated Approach boasts superior accuracy; however, it requires precise measurements of varying aircraft/engine types under disparate loadings, trajectories, and meteorological conditions, and the deployment measurements of thrust reversal under distinct meteorological scenarios for varying aircraft/engine configurations—data that are often times elusive in the public domain. Given these considerations, this study judiciously employs the Advanced Approach, striking a balance between precision and data accessibility. The Advanced Approach of ICAO defines the landing and take-off cycle as all aircraft operations conducted below 3000 ft (equivalent to 915 m) near the airport. These operations encompass four distinct modes of operation (take-off, climb, approach, and idle), each involving a specific thrust setting and a time-in-mode [36]. Followed by the ICAO Advanced Approach, this paper quantifies LTO cycle emissions as Equation (1):

$$E_{i,LTO} = \sum \left( TIM_{jk} \times 60 \right) \times \left( \frac{EF_{jk}}{1000} \right) \times \left( EI_{jk} \right) \times NE_j \tag{1}$$

where

$E_{i,LTO}$ is the Total emissions of pollutant *i* in the LTO cycle.
$TIM_{jk}$ is the Working time for mode *k* (take-off, climb, approach, and idle).
$EF_{jk}$ is the Fuel flow for mode *k* (take-off, climb, approach, and idle) in each engine used on aircraft type *j*.
$EI_{jk}$ is the Emission indices for the pollutant of the engine used on aircraft type *j* in mode *k* (take-off, climb, approach, and idle).
$NE_j$ is the Number of engines used on aircraft type *j*.

The engine thrust settings and durations for each flight mode in the standard LTO cycle are presented in Table 3.

**Table 3.** ICAO's thrust settings and duration for LTO cycle.

| Mode | Thrust Setting | Duration Time (Min) |
|---|---|---|
| take off | 100% | 0.7 (42 s) |
| climb | 85% | 2.2 (132 s) |
| approach | 30% | 4 (240 s) |
| idle | 7% | 26 (1560 s) |

### 3.2.2. Emissions during the Cruise, Climb, and Descent (CCD) Flight Cycle

BADA's emissions data, predicated upon specific meteorological conditions and flight altitudes, compute for various flight distances (or equivalently, flight times) that range from 125 nautical miles to a considerable 8000 nautical miles, corresponding to flight times spanning from a brief 19 min to an extensive 1032 min. In this study, a method of linear interpolation calculation [21] is employed to quantify the CCD emissions for instances where flight distances and times of the study diverge from those simulated within BADA, as demonstrated in the following formula:

$$E_{i,CCD} = \frac{E_{i,up} - E_{i,low}}{D_{i,up} - D_{i,low}} \left( D_{i,up} - D_{i,low} \right) + E_{i,low} \tag{2}$$

where $I$ represents the categorization of atmospheric pollutants (HC, CO, NO$_x$, and CO$_2$). $D$ represents the real flight time during the CCD cycle stage for a specific flight, $D_{i,up}$ is the BADA test time that approaches flight time $D$ but remains slightly inferior, and $D_{i,low}$ is the BADA test time nearest to $D$, but superior. The variables $E_{i,low}$ and $E_{i,up}$ denote the pollutant emissions corresponding to the flight times $D_{i,low}$ and $D_{i,up}$, respectively.

In conclusion, the total emissions ($E_{Total}$) for a complete flight can be estimated by adding both $E_{i,LTO}$ and $E_{i,CCD}$:

$$E_{Total} = E_{i,LTO} + E_{i,CCD} \tag{3}$$

### 3.2.3. Carbon Dioxide Equivalent Calculation

Carbon dioxide equivalent ($CO_{2e}$) is used to quantify the commensurate effect on global warming that a given amount and type of greenhouse gas may engender. Herein, the equivalent factor for $CO_2$ is set as 1, while for HC, CO, and NO$_x$, the respective equivalent factors are 1.57, 84, and 298 [37]. Consequently, the calculation of the carbon dioxide equivalent of civil aviation emissions can be facilitated using the following equation:

$$E_{CO_{2e}} = E_{CO_2} + 1.57 \times E_{CO} + 84 \times E_{HC} + 298 \times E_{CO_X} \tag{4}$$

where $E_{CO2e}$ represents the carbon dioxide equivalent emission; $E_{CO2}$, $E_{CO}$, $E_{HC}$, and $E_{NOx}$ denote $CO_2$ emissions, CO, HC, and NO$_x$ emissions, respectively.

### 3.2.4. Carbon Emissions Allocation between Cities

The present study employs a spatial allocation method to allocate carbon emissions from an individual flight during both the LTO cycle and the CCD cycle to the respective departure and arrival airports. Specifically, the carbon emissions generated during thetake-off and climb phases are attributed to the departure airport, while those accrued during the approach and idle phases are assigned to the arrival airport. The emissions during the CCD cycle are evenly apportioned between the departure and arrival airports [21]. The calculations are as follows:

$$E_{departure} = E_{CO2e,T} + E_{CO2e,C} + E_{i,CCD}/2 \tag{5}$$

$$E_{arrival} = E_{CO2e,A} + E_{CO2e,I} + E_{i,CCD}/2 \qquad (6)$$

where $E_{departure}$ and $E_{arrival}$ represent the carbon emissions of the departure airport and the arrival airport, respectively; $E_{CO2e,T}$, $E_{CO2e,C}$, $E_{CO2e,A}$, and $E_{CO2e,I}$ denote carbon emissions from the take-off, the climb, the approach, and the idle phase, respectively; $E_{i,CCD}$ represents the emissions during the CCD cycle of the flight.

Building on this methodology, the sum of carbon emissions from an airport over a specified time period is the aggregate emissions of all departing and arriving flights at that airport. Some cities may have more than one airport, so the emissions for each city are the combined emissions of all its airports.

According to the gravity model in the international trade theory, the trade volume between two countries is influenced by their respective economic sizes (usually represented by GDP) and the distance between them. In this context, we aim to use population size as a substitute for GDP to allocate aviation carbon emissions between two cities.

## 4. Results

### 4.1. Changes in National Carbon Emissions

(1) Fluctuations in Emissions of Various Pollutants

Utilizing the ICAO Advanced Approach, we developed a computational model. By integrating comprehensive flight data and engine specifications, the model calculates the carbon emissions of each individual flight in China for the period 2012 to 2021. The emissions under consideration include $CO_2$, CO, HC, and $NO_x$. Employing Equation (4), these emissions were converted into carbon dioxide equivalents ($CO_{2e}$), with the results detailed in Figure 3.

As shown in Figure 3, between 2012 and 2019, China experienced a significant increase in civil aviation carbon emissions, with the annual growth rate of carbon dioxide equivalents reaching 12.51%. This peaked at 186.53 million tons in 2019, indicating a period of rapid growth for China's aviation industry. Upon evaluating the growth dynamics, it is evident that different pollutants exhibit distinct growth patterns. Predominant pollutants showcase accelerated growth velocities. Specifically, $CO_2$ and $NO_x$ have average annual emissions of 58,175 thousand tons and 328.32 thousand tons, respectively. Conversely, HC and CO, characterized by their relatively modest emission volumes, register annual emissions of 9.23 thousand tons and 72.24 MT, respectively. In terms of growth kinetics, $CO_2$ stands out with the most pronounced growth, clocking in at 12.76%, while HC lags, recording an annual growth of merely 2.52%. In a proportional context, among all greenhouse-related emissions, $CO_2$ prevails with a significant dominance, accounting for an average of 80.62% of the total emissions. In contrast, HC contributed minimally, accounting for only 0.10%. The period between 2019 and 2021, however, marked a deviation due to the COVID-19 pandemic. As a result, there was a sharp decrease in aviation carbon emissions starting in 2020, attributable to the pandemic-induced urban lockdowns. The stringent measures implemented by the Chinese government led to a dramatic reduction in 2020, with a more moderate decrease observed in 2021. Such data suggest that significant external events, like the COVID-19 pandemic, can result in short-term reductions in aviation carbon emissions. However, these events are unlikely to change the long-term upward trend of emissions.

(2) Emission changes during LTO and CCD cycles

To assess the variations in aviation pollutant emissions, we disaggregated the total emissions from 2012–2021 into the LTO and CCD phases, as depicted in Figure 4.

Figure 4 reveals distinct characteristics of various pollutants during the LTO and CCD phases. In terms of absolute emissions, HC, CO, and $NO_X$ are significantly emitted during the LTO phase, with minimal emissions during the CCD phase. Conversely, $CO_2$ exhibits an inverse trend, with a smaller proportion of emissions during the LTO phase and predominant emissions in the CCD phase. The emission percentages of HC, $NO_X$, CO,

and $CO_2$ during a single CCD phase, in relation to their total phase emissions, are 0.77%, 2.08%, 0.49%, and 96.24%, respectively. This discrepancy arises primarily because, during the LTO phase, engines operate at lower rotational speeds. In contrast, during the CCD cruise phase, engines work at an optimal and efficient speed with a favorable fuel-to-air mixture, ensuring high combustion efficiency. Consequently, the LTO phase displays higher emissions of HC, $NO_X$, and CO, whereas the CCD phase is characterized by greater $CO_2$ emissions.

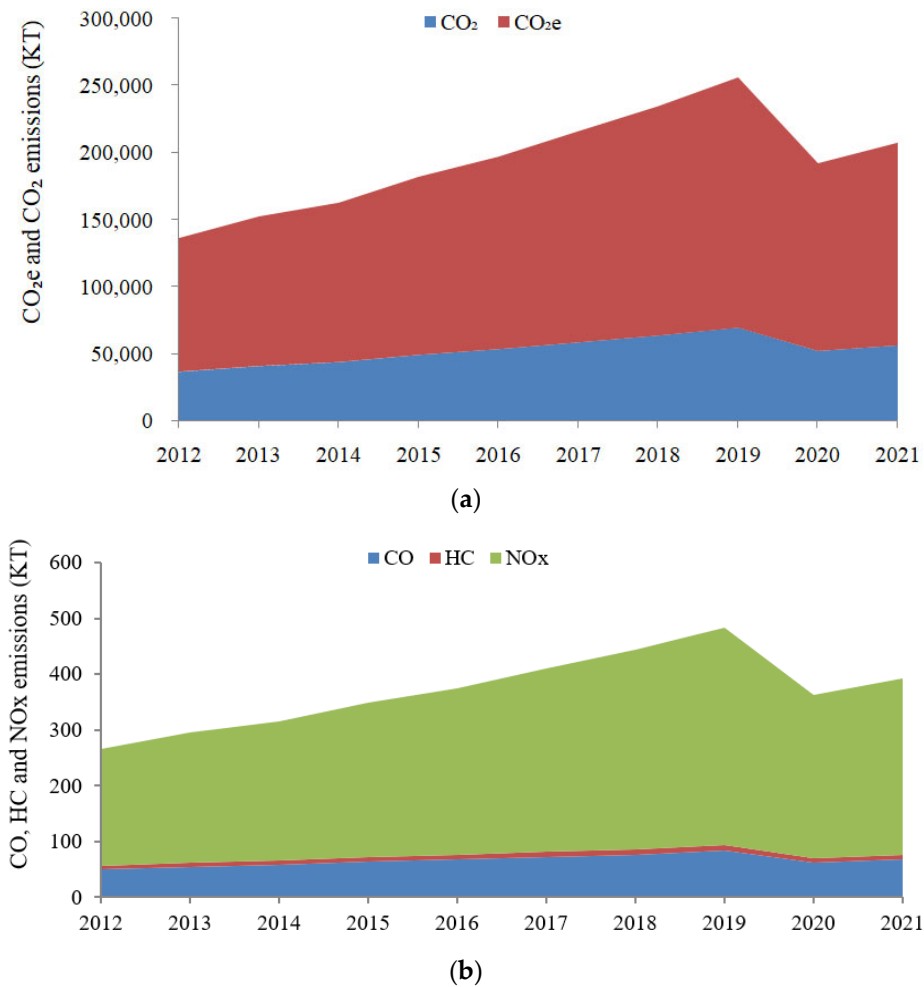

**Figure 3.** Annual emission of various pollutants from 2012 to 2021 in China. (**a**) $CO_{2e}$ and $CO_2$ emissions from 2012 to 2022. (**b**) CO, NOx, and HC emissions from 2012 to 2021.

Additionally, we calculated the annual growth rates of each pollutant's emissions during the LTO and CCD phases from 2012 to 2021, as depicted in Figure 5. The annual emission growth rates for all pollutants during the CCD phase surpassed those of the LTO phase. Notably, $CO_2$ exhibited the highest growth rate during the CCD phase at 21.44%. Meanwhile, the LTO phase saw $NO_X$ as the fastest-growing pollutant with a growth rate of 13.78%. Amidst a universal increase in pollutant emissions, the carbon dioxide equivalent emissions grew by 13.64% and 14.81% during the LTO and CCD phases, respectively.

### 4.2. Changes in Megacity Carbon Emissions

(1) Temporal variations in pollutant emission for China's megacities

In this section, emissions from takeoff and ascent during the aircraft's LTO phase are ascribed to the departure city, while those from approach and idling are assigned to the destination city. Emissions from the CCD cycle phase are then spatially distributed to the

cities where each airport is located. By aggregating the emissions from both phases, we obtain the total civil aviation flight carbon emissions for each city. As depicted in Figure 6, the stark regional disparities are evident in China's aviation-related carbon emissions. Nineteen megacities contribute to over half of the nation's total carbon emissions. Between 2012 and 2021, the quantified emissions for CO, HC, and $NO_X$ from China's civil aviation sector amounted to 722.09, 92.25, and 3282.12 thousand tons, respectively. Notably, the $CO_2$ and carbon dioxide equivalent were staggering 581.55 and 1568.51 million tons. Among these, the emissions of the said pollutants—CO, HC, $NO_X$, and $CO_2$—in the 19 megacities comprised 58.86%, 57.38%, 62.83%, and 61.81% of the total, respectively. Furthermore, the $CO_2$ equivalent emissions from these cities accounted for an impressive 62.08% of the total (refer to Figure 6 for a detailed representation). Remarkably, out of the 270 sampled cities, 207 cities reported $CO_2$ emissions of less than 1000 thousand tons. Cumulatively, their contribution is a mere 9.29% of the total carbon emissions from all cities.

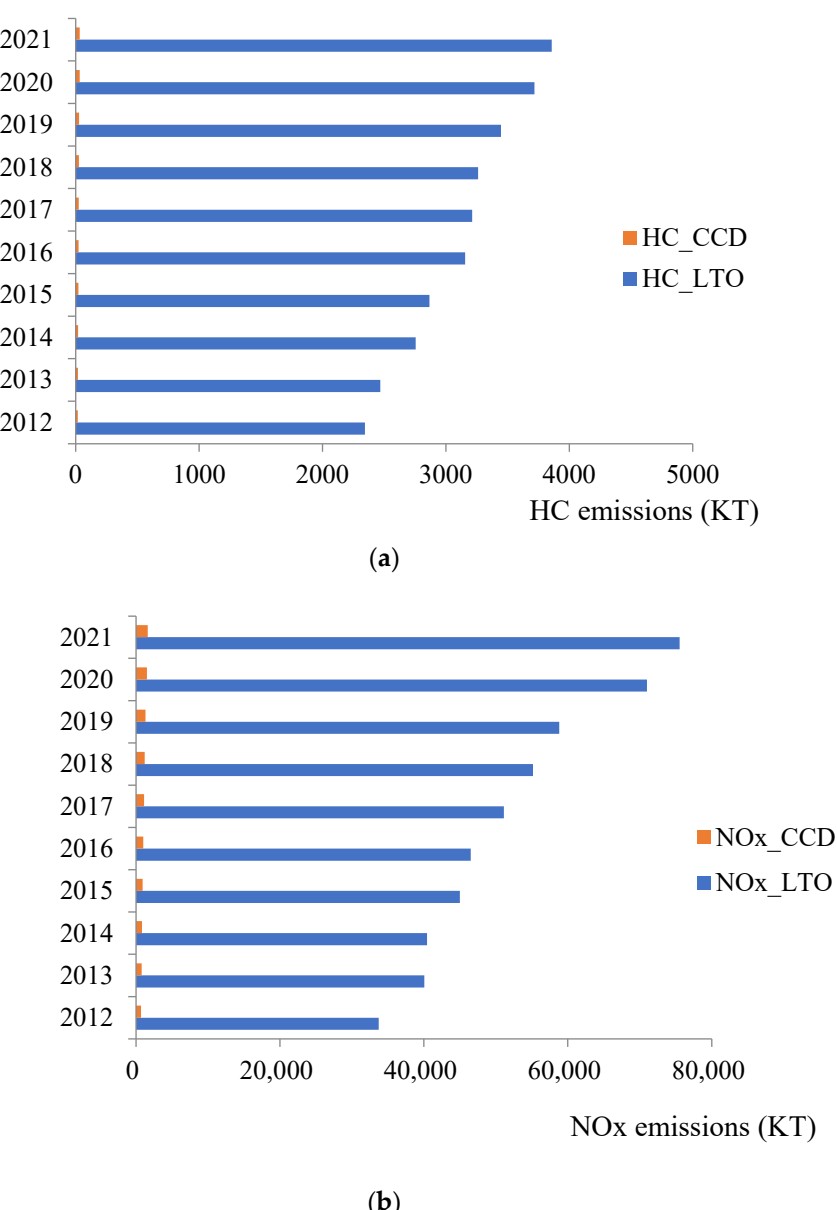

**Figure 4.** *Cont.*

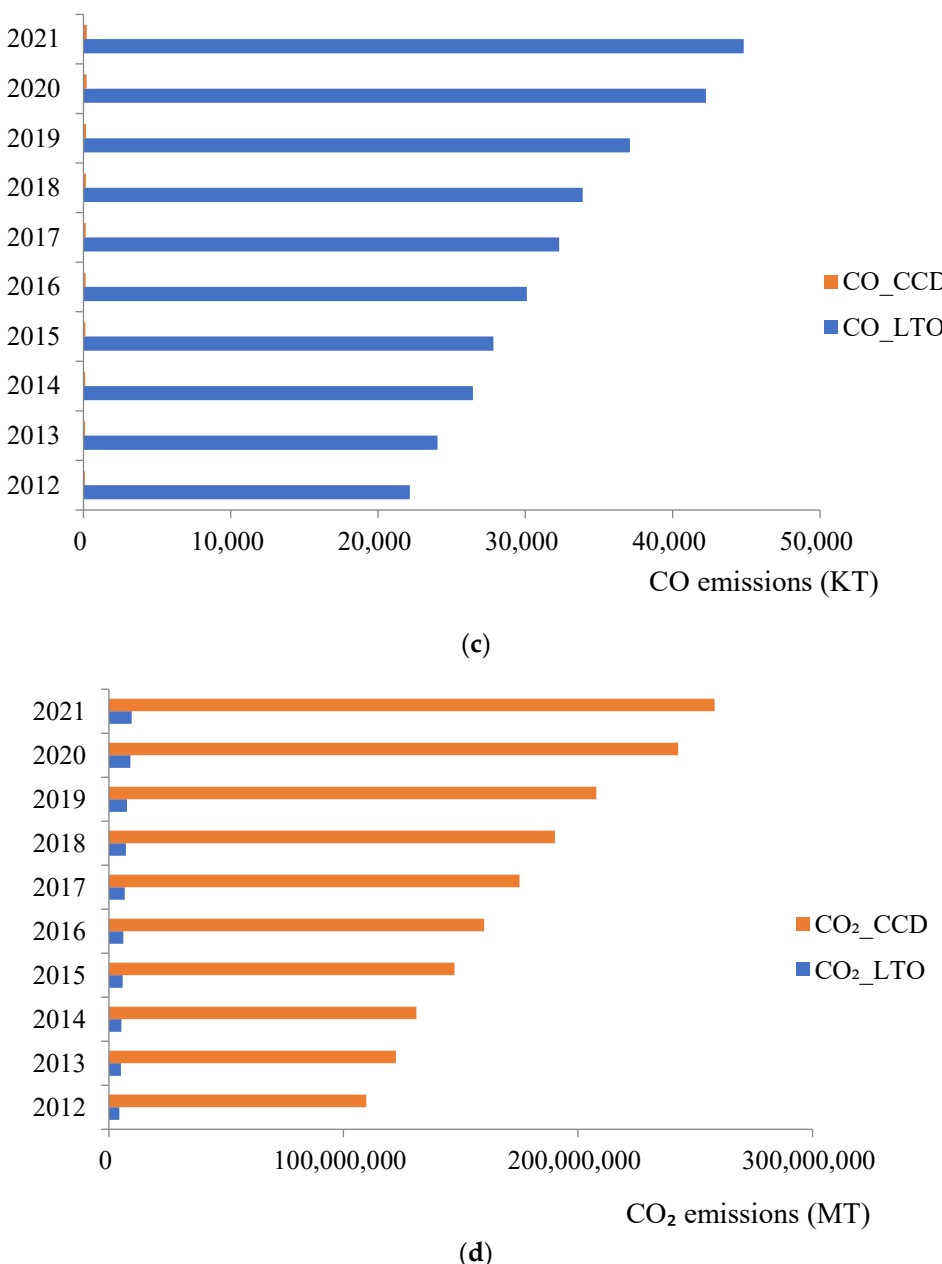

**Figure 4.** LTO and CCD emission of various pollutants from 2012 to 2021. (**a**) HC emissions in LTO and CCD cycles from 2012 to 2021. (**b**) $NO_X$ emissions in LTO and CCD cycles from 2012 to 2021. (**c**) CO emissions in LTO and CCD cycles from 2012 to 2021. (**d**) $CO_2$ emissions in LTO and CCD cycles from 2012 to 2021.

It is imperative to highlight that there is considerable heterogeneity in aviation carbon emissions even within these megacities. Figure 7 depicts the average carbon emission of the 19 largest cities, categorizing them into approximately four distinct tiers based on their emission volumes. Beijing and Shanghai, topping the list with flight volumes, record $CO_2$ emissions at 55.89 MT and 44.24 MT, respectively, anchoring them solidly in the highest emission bracket. Cities such as Guangzhou, Chengdu, and Shenzhen follow closely, with emissions varying between 24.99 MT and 31.30 MT, positioning them in the second echelon. Kunming, Xi'an, Chongqing, Hangzhou, Nanjing, and Zhengzhou fall within a range of 12.60 MT to 21.01 MT, situating them in the third tier of aviation $CO_2$ emissions. Cumulatively, the cities spanning these first three tiers account for a substantial 44.61% of China's total aviation $CO_2$ emissions. Lastly, cities including Harbin, Tianjin, Changsha,

Wuhan, Qingdao, Shenyang, Dalian, and Jinan each have $CO_2$ emissions around and below the 10 MT threshold, placing them in the fourth tier. It is noteworthy that the aviation $CO_2$ emissions of cities in the first three tiers are between 2 and 8 times that of Dalian, which is positioned last and emitted 7.21 MT. This underscores a significant imbalance in aviation $CO_2$ emissions among cities, even within the subset of mega-cities. The top 10 emitting mega-cities contributed 46.87% of the national aviation $CO_2$ emissions and 75.83% of the total 19 megacities' emissions.

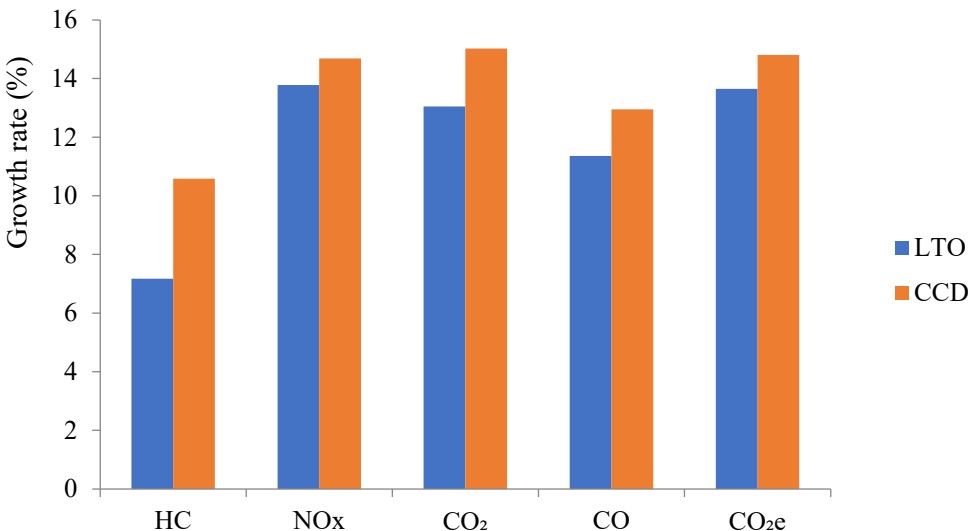

**Figure 5.** Annual growth rates of various pollutants emissions in LTO and CCD cycles.

Further, as illustrated in Figure 7, the emission trajectories of various pollutants within these cities generally parallel one another. Cities with higher $CO_2$ emissions concurrently exhibit elevated emissions of HC, $NO_X$, and CO. Among the cataloged pollutants, $CO_2$ emissions are the most abundant, followed by $NO_X$, with HC registering as the least prevalent.

Figure 8 presents the $CO_2$ emission intensities for the 19 megacities, obtained by dividing the city-specific $CO_2$ emissions by their respective flight frequencies. Notably, there is not necessarily a direct correlation between the magnitude of $CO_2$ emission intensities and the volume of emissions. For instance, Harbin and Shenyang rank 12th and 17th, respectively, in terms of $CO_2$ emissions among these cities, yet they stand at 6th and 7th positions for emission intensities. The optimization of flight route designs and enhancements in ground operations at airports for cities with elevated emission intensities merit earnest consideration in shaping the future trajectory of China's aviation sector.

(2)    The historical evolution of aviation carbon emissions in Chinese cities (2012–2021)

From 2012 to 2021, the pattern of urban aviation carbon emissions in China displayed a relative consistency (See Figure 9). The cities registering the highest emissions during this period, ranked in descending order, were Beijing, Shanghai, Guangzhou, Shenzhen, Chengdu, Kunming, Xi'an, Chongqing, Hangzhou, and Urumqi. While there was slight volatility in the ranking order of individual cities over the decade, the list of top 10 emitters remained unaltered. Notably, the top quartile—comprising Beijing, Shanghai, Guangzhou, and Shenzhen—maintained their respective positions throughout the ten years. However, when examining growth rates in emissions, several tourist cities exhibited staggering surges in aviation carbon outputs. A deeper analysis of the emissions growth rate reveals an astonishing surge in certain tourist-rich cities. Emission rates in touristic cities of Xinjiang and Tibet, including Aksu, Hami, Ali, Korla, Yushu, and Nyingchi, surged between 200 and 2000%. Additionally, Hanzhong (Shaanxi), Libo (Guizhou), and Panzhihua (Sichuan)

experienced remarkable increases, registering growth rates of 4827%, 2026%, and 1538%, respectively.

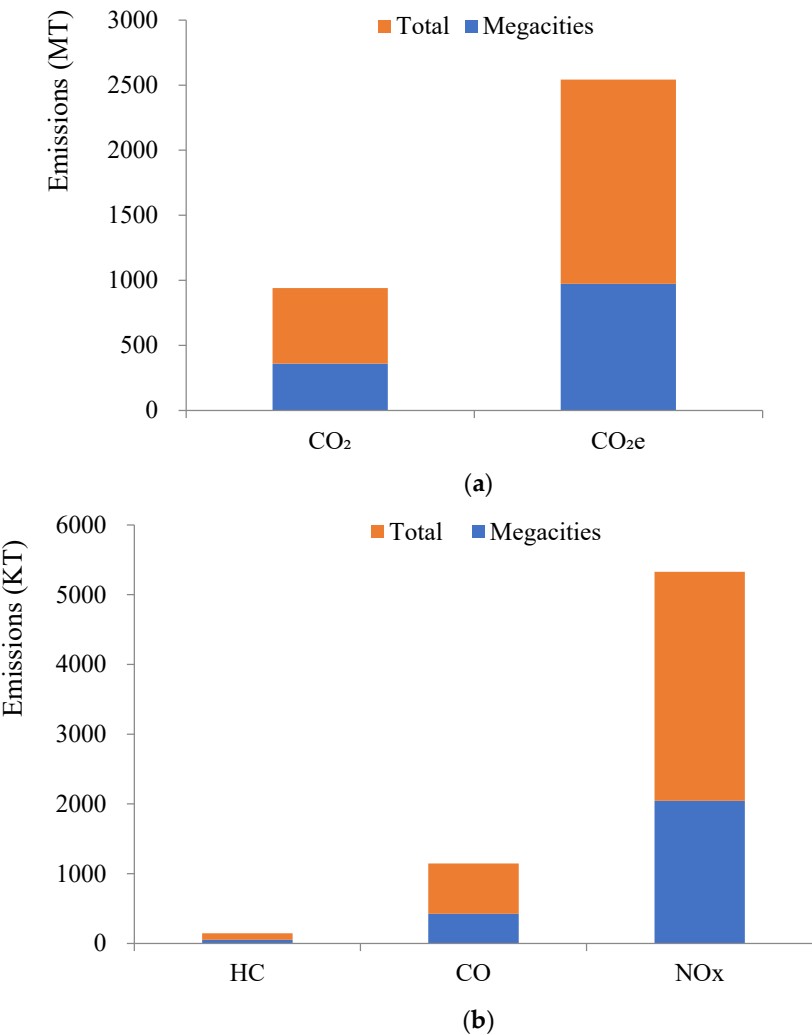

(**a**)

(**b**)

**Figure 6.** Comparison of various pollutants emissions between China's megacities and the total. (**a**) Comparison of $CO_2$ and $CO_{2e}$ emissions between China's megacities and the total. (**b**) Comparison of HC, CO, and NOx emissions between China's megacities and the total.

Recent geopolitical and economic initiatives, such as the Belt and Road Initiative and the accelerated establishment of the Chengdu-Chongqing Economic Circle, appear to have provided impetus to the aviation industry in western provinces, leading to a rapid increase in $CO_2$ emissions from aviation. In metropolitan areas, cities such as Nanjing, Shenyang, Qingdao, and Xi'an, which have witnessed fast-paced economic development and urbanization, have seen their aerial carbon emissions rise by 101.18%, 100.60%, 100.28%, and 98.65%, respectively. As these cities progressed economically and demographically, the demand for aviation services grew commensurately. Specific cities like Kunming and Urumqi, endowed with unique geographical and cultural attributes, have seen a spike in both domestic and international tourism, thereby amplifying their aviation-related emissions. The infrastructural evolution in Hangzhou, Chongqing, and Chengdu, characterized by the introduction of new domestic and international flight routes and the establishment or planning of aviation hubs, has further exacerbated emission levels. Contrastingly, major hubs such as Beijing, Guangzhou, and Shanghai recorded modest growth rates of 35.94%, 42.02%, and 37.77%, respectively. Given their foundational role in China's economic and transportation matrix, the relative growth was bound to be subdued. Furthermore, height-

ened environmental considerations in these cities have likely led to the adoption of carbon mitigation policies and initiatives. Except for the years 2020 and 2021, which were anomalistic due to the pandemic, only a few cities recorded a decrease in aerial carbon emissions: Beijing in 2016, Kunming in 2018, and Jinan in 2014. The broader implications of these findings warrant a deeper exploration into urban policies, infrastructural development, and the interplay of economic initiatives with environmental outcomes.

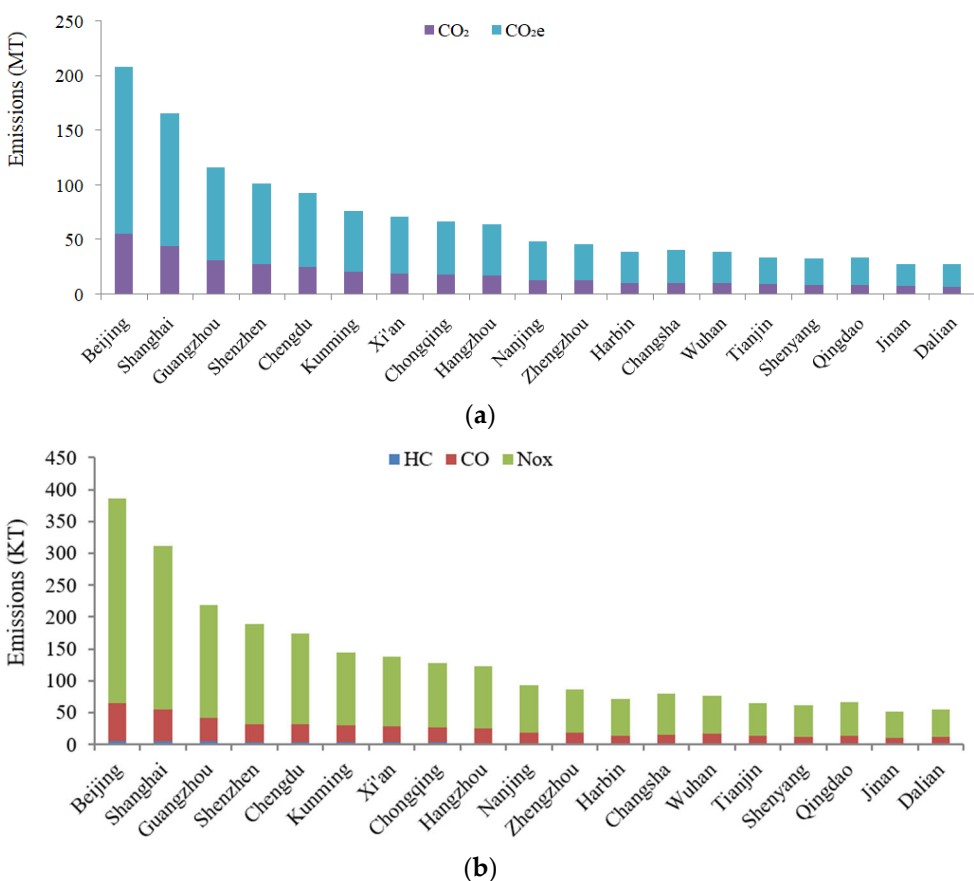

(**a**)

(**b**)

**Figure 7.** Various pollutants emissions in China's megacities (2012–2021). (**a**) Comparison of $CO_2$ and $CO_{2e}$ emissions. (**b**) HC, CO, and NOx emissions.

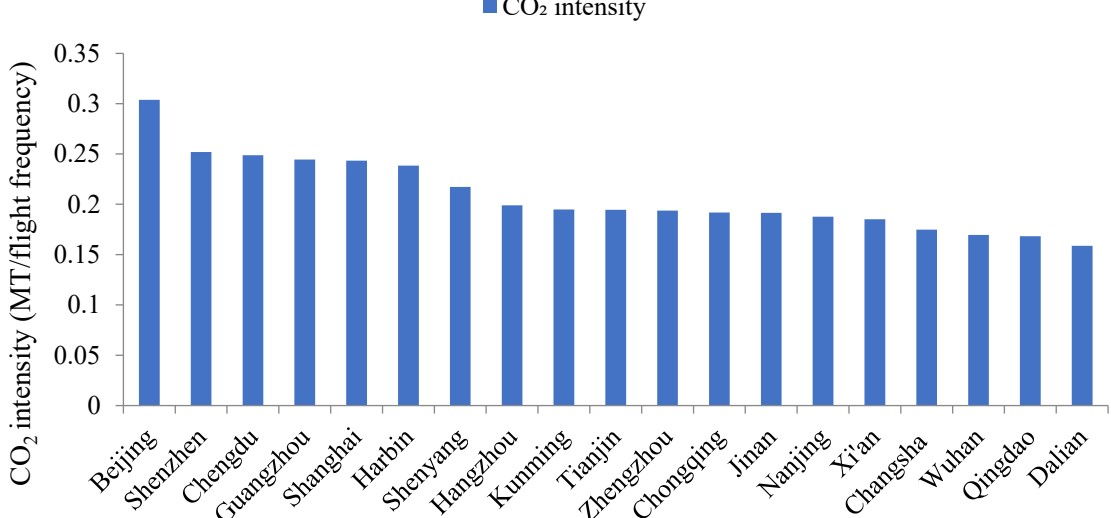

**Figure 8.** Aviation $CO_2$ emission intensities of China's megacities.

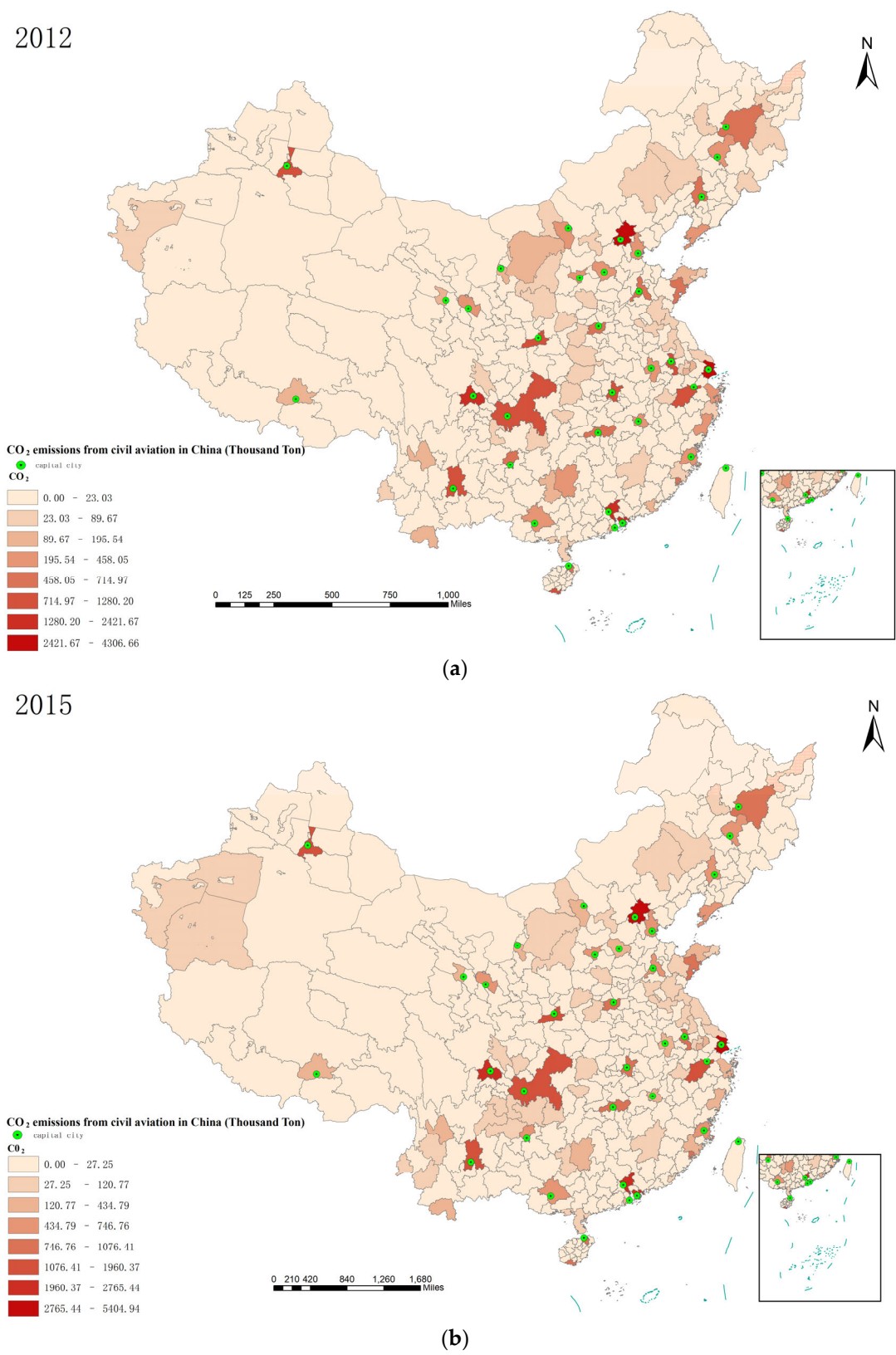

**Figure 9.** *Cont.*

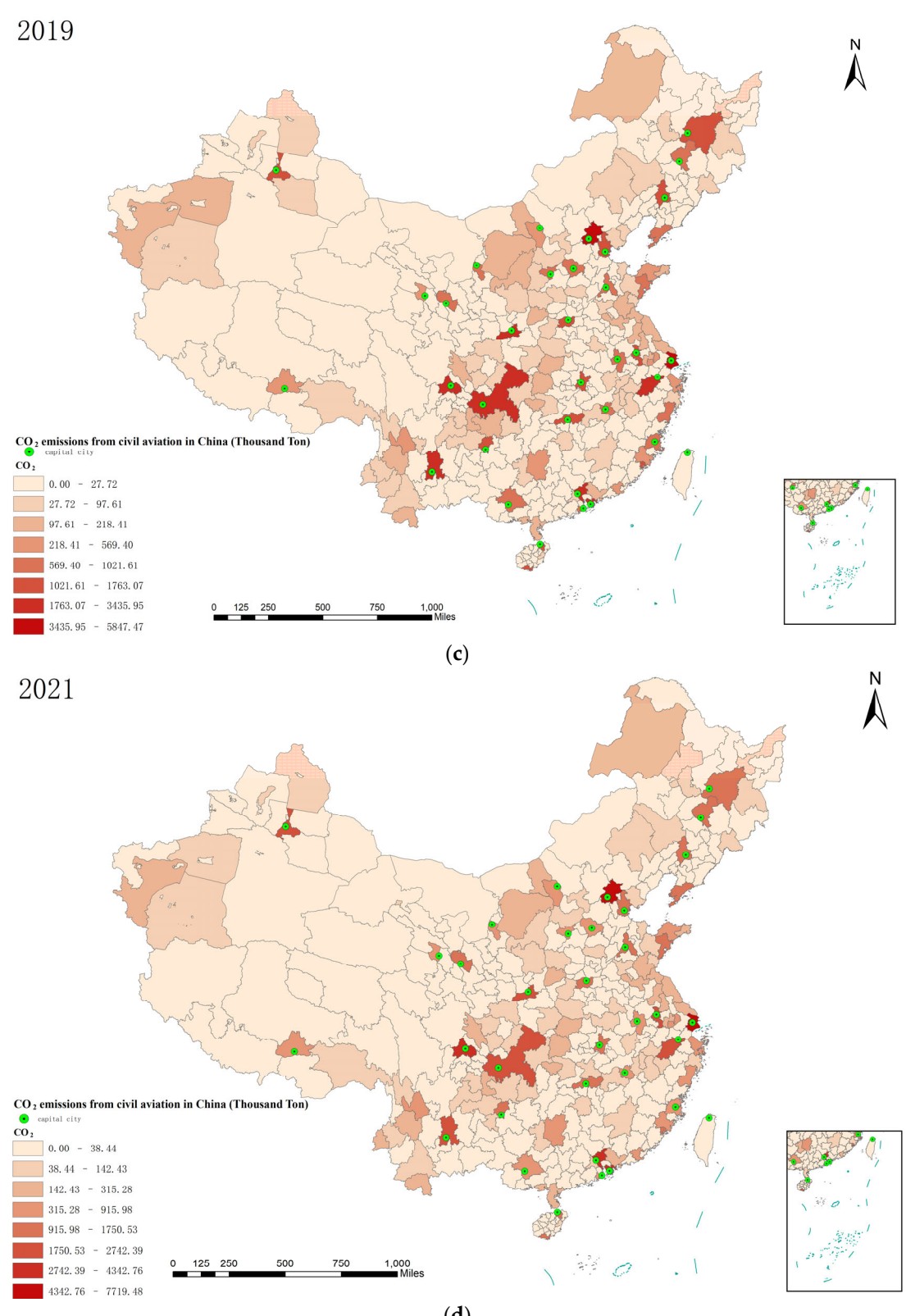

**Figure 9.** Historical evolutions of civil aviation $CO_2$ emissions in China. (**a**) Civil aviation $CO_2$ emissions in 2012. (**b**) Civil aviation $CO_2$ emissions in 2015. (**c**) Civil aviation $CO_2$ emissions in 2019. (**d**) Civil aviation $CO_2$ emissions in 2021.

*4.3. Carbon Emissions from Different Aircraft Types*

To quantify the carbon emissions attributed to different aircraft models, cumulative emissions for each flight, segmented by aircraft type, were computed. Table 2 presents the emissions for the ten aircraft models with the highest cumulative emissions. As shown in Table 2, of these top ten emitters, the Embraer E90 manifests the lowest per-flight carbon emissions, averaging 1490.93 kg per departure. In contrast, the Airbus A332 records the highest, with an average of 9065.02 kg per flight.

In Section 2.4, we found that the most common aircraft models for China's domestic flights are the Airbus A320 and Boeing B738, making up 35.36% and 27.10% of the total flights, respectively. Notably, these models are also the main contributors to carbon emissions, with their $CO_2$ equivalent emissions accounting for 31.98% and 23.4% of the total, respectively. Both the A320 and B738, typical for domestic routes in China, are twin-engine, single-aisle narrow-body jets with seating capacities ranging from 140 to 189. While other models have similar configurations and sizes, the per-flight carbon emissions of the B738 are slightly lower than the A320. Following them, the Airbus A321, A319, and Boeing 737 are next in line in terms of emissions.

When considering flight frequency, the top emitters are the A332, A330, and A333 models, as shown in Table 4. A possible reason is the nature of these aircraft. Although the A332, A330, and A333 might have fewer flights than the A320 and B738, their wide-body design suggests they have larger fuel capacities and faster consumption rates. Additionally, they often fly long-haul routes, which means more fuel is burned, resulting in higher $CO_2$ equivalent emissions. In contrast, the A320 and B738, primarily used for medium-to-short-haul routes, may not cover as much total distance as long-haul aircraft. Thus, even with fewer flights, they might emit more carbon per flight. Domestic flights, with their frequent take-offs and landings, differ from long-haul international flights that have extended cruising periods. Since the take-off phase uses the most fuel, this pattern can lead to increased emissions.

**Table 4.** Top 10 aircraft models by carbon emissions (2012–2021).

| Airplane | No. of Seat | $CO_{2e}$ (KT) | Percent (%) | Flight (Frequency) | KG/Flight |
|---|---|---|---|---|---|
| A320 | 140–170 | 48,199.21 | 0.32 | 11,101,090 | 4341.84 |
| B738 | 162–189 | 35,270.38 | 0.23 | 14,481,012 | 2435.63 |
| A321 | 185–240 | 12,061.35 | 0.08 | 3,038,984 | 3968.87 |
| A319 | 124–156 | 9388.27 | 0.06 | 4,255,836 | 2205.98 |
| B737 | 102–189 | 6261.80 | 0.04 | 3,193,190 | 5122.15 |
| A330 | 230–440 | 3516.33 | 0.02 | 459,862 | 7646.48 |
| A333 | 277–440 | 3414.06 | 0.02 | 387,478 | 8810.97 |
| B73F | 96–114 | 2766.61 | 0.02 | 397,020 | 6968.43 |
| E90 | 230–406 | 2196.19 | 0.01 | 1,473,030 | 1490.93 |
| A332 | 305–550 | 2026.23 | 0.01 | 223,522 | 9065.02 |

## 5. Conclusions and Implications

### 5.1. Conclusions

With China's ongoing urbanization and industrialization, the growth of its megacities has become a clear trend. This study provides a comprehensive analysis of the trends and characteristics of civil aviation carbon emissions in these megacities. Initially, we collected flight data for China's main domestic routes, including aircraft types, flight frequencies, routes, distances, and flight durations. Next, we calculated the total emissions of carbon dioxide ($CO_2$), carbon monoxide (CO), hydrocarbons (HC), and nitrogen oxides (NOx), as well as the $CO_2$ equivalent for each airline and route. This includes emissions during both the Cruise Climb Descent (CCD) and Landing Take-Off (LTO) phases, with the latter calculated using the ICAO Advanced methodology and the former using the approach by Eskenazi (2022) [21]. This method of tracking emissions provides a clearer understanding

of the environmental impact of flight activities in China's megacities, offering key data to develop targeted counter measures. Our findings include:

(1) Even though China is an active participant in international aviation communities and has pledged to continue with the CORSIA carbon reduction plan, the data show that the carbon emissions from China's aviation sector have been increasing, rising by 12.52% from 2012 to 2022.

(2) Of all the pollutants emitted by aircraft, $CO_2$ is the most dominant, making up 83.87% of total emissions. Importantly, HC, $NO_x$, and CO are mainly released during the LTO phase, while $CO_2$ emissions occur mostly during the CCD phase, accounting for 96.24%. $CO_2$'s rapid growth in the CCD phase and $NO_x$'s increase during the LTO cycle are major contributors to the aviation-induced greenhouse effects and should be the primary focus of carbon reduction efforts.

(3) There's a clear unevenness in carbon emissions across cities, with 19 megacities producing a significant 62.08% of total $CO_2$ equivalent emissions. In comparison, among the total 270 cities, the combined emissions from the 207 least-emitting cities make up only 9.29% of the nation's total.

(4) Cities with high emission rates include major aviation hubs like Beijing, Shanghai, Guangzhou, and Shenzhen, but also cities like Harbin and Shenyang, which, despite their lower total emissions, have high emission rates.

(5) The pattern of urban carbon emissions is changing. Cities in the west, such as Xinjiang, Tibet, Shaanxi, and Guizhou, have seen rapid increases in their emissions, with some growing by as much as 4827%. Among the megacities, growth rates vary, with traditional hubs like Beijing, Shanghai, and Guangzhou showing slower growth compared to cities like Nanjing, Shenyang, Qingdao, Xi'an, Hangzhou, Chengdu, Chongqing, Kunming, and Shenzhen. Urumqi's emissions have also risen by 85.39%, driven by its growing popularity as a tourist destination.

It is noteworthy that our findings align closely with the results presented by Liu et al. (2019) [24]. They employed a bottom-up approach, as outlined by the ICAO, for emissions during the LTO phase. However, due to data constraints in the CCD phase, they opted to compute total aircraft carbon emissions based on annual kerosene consumption and then subtracted emissions from the LTO phase. Their results indicated that in 2015, emissions for HC stood at 4.77 KT, CO at 59.63 KT, NOx at 304.77 KT, and $CO_2$ at 59,961 KT. In contrast, our computations revealed emissions for HC at 8.32 KT, CO at 62.83 KT, NOx at 277.61 KT, and $CO_2$ at 49,141 KT. Our methodology, which takes into account the aircraft model and engine for each individual flight during both the LTO and CCD phases, provides a more precise estimation of airborne pollutant emissions.

*5.2. Implications*

Drawing from the aforementioned findings, it is evident that imbalances exist both within China's urban aviation carbon emissions and among various pollutants. Traditional metropolises such as Beijing, Shanghai, and Guangzhou have contributed significantly to aviation carbon emissions. However, cities in the western region like Chengdu, Kunming, and Chongqing have seen a marked increase, while in the east, cities like Harbin and Shenyang exhibit particularly high carbon emission intensities. Notably, greenhouse gases such as $CO_2$ and $NO_X$ have seen significant growth, dominating the emission spectrum. Such trends pose substantial challenges for China's aviation sector to peak and neutralize its carbon footprint. This study offers targeted guidance for emission reductions within the aviation industry. Specifically, the following measures are proposed:

(1) Encourage Research and Development in Aviation Fuel Technology. Given the dominant role of $CO_2$ in emissions, it is essential to prioritize and accelerate the development and utilization of more efficient and eco-friendly alternative aviation fuels (SAF). Additionally, the promotion and refinement of aviation carbon accounting and monitoring techniques are crucial to ensure precise carbon emission tracking across cities and flight routes.

(2)   Intensify Management of Pollutants during the LTO Phase. Emissions of $NO_X$, HC, and CO are notably high during the LTO phase. This necessitates stricter monitoring and management measures, such as upgrading air traffic control systems and minimizing aircraft hover time in the air.

(3)   Strengthen Carbon Emission Management in Megacities: Given the high proportion of $CO_2$ emissions from megacities, it is vital to implement aviation carbon emission targets tailored for these metropolises.

(4)   Support Green Aviation Development in Western Cities: Considering the rapid growth rate of carbon emissions in western cities, efforts should be focused on fostering their transition to green aviation, advocating for low-carbon technologies, and steering the trajectory of sustainable aviation practices.

(5)   Optimize Air Route Designs for High-Emission Intensity Airports: For airports like those in Harbin and Shenyang with elevated emission intensities, strategies should involve precise flight path planning to effectively minimize flight duration and distance. Ground operations at airports should also be optimized, emphasizing improved ground services, reduced taxiing durations, and minimizing unnecessary engine operations.

(6)   Implement Differentiated Aviation Carbon Emission Control Strategies: Based on the varying carbon emission growth rates across cities, differentiated control strategies are necessary. Cities with higher growth rates should face more stringent emission control measures.

This polished translation captures the academic essence reminiscent of journals like the American Economic Review, maintaining clarity and elegance in presenting the research findings and recommendations.

In summary, there's a clear imbalance in China's aviation carbon emissions, both among cities and across different pollutants. While megacities like Beijing, Shanghai, and Guangzhou have been big contributors, emerging centers in the west and cities in the east like Harbin and Shenyang have shown notable growth or high emission rates. Given the large increases in $CO_2$ and NOx, which have strong greenhouse effects, China's aviation sector faces significant challenges in meeting its carbon goals. This study offers a roadmap for targeted reductions in aviation emissions.

*5.3. Limitations and Future Research Directions*

Employing methods from the ICAO and Eskenazi (2022), we calculated civil aviation carbon emissions from 2012 to 2021 in China, with a focus on carbon emissions in mega-cities with populations over 5 million. Yet, our study has limitations. First, when quantifying emissions during the CCD phase, we need to match aircraft models to their engines. However, the ICAO database has a limited selection of engine models. Therefore, when a direct match was not available, we used similar models. For example, the B73G was replaced with the B737-700, and the general A330 was substituted with the A330-200. Secondly, given the absence of passenger and mileage data, our calculations could not take into account aircraft payload or render a more scientific computation of emissions per kilometer flown. Instead, we only estimated emissions per individual flight. As more data become available, it would be useful to measure carbon emissions for each flight, taking into account the aircraft's payload. In conclusion, forthcoming research could look into the effects of environmental policies, the rise of sustainable aviation fuels, and developments in hydrogen energy on aviation's carbon emissions. An exploration into the network dynamics of aviation carbon emissions would also be of interest.

**Author Contributions:** Y.S. conceived the research idea and wrote the manuscript; Y.D. conducted the analysis; M.C. was in charge of charting. All authors have read and agreed to the published version of the manuscript.

**Funding:** This research is financially supported by the National Social Science Fund of China (23BJL096), Jiangxi Social Science Planning Fund (No. 23JL06), Jiangxi University Humanities and Social Sciences Project Fund (No. JJ20112), Jiangxi Province Degree and Graduate Education Teaching Reform Research Project (JXYJG-2021-149), and Key R&D Projects of Jiangxi Provincial Department of Science and Technology: No. 20192BAA208014.

**Data Availability Statement:** The data used in this study are all public data and have been mentioned in detail in the data description.

**Acknowledgments:** We would also like to thank Jia Ling from Nanchang Hangkong University and Zhang Tian from Nanjing University of Aeronautics and Astronautics for their assistance in data collection and cleaning.

**Conflicts of Interest:** The authors declare no conflict of interest.

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
