# Peer review of "From Takeoff to Touchdown: A Decade’s Review of Carbon Emissions from Civil Aviation in China’s Expanding Megacities"

_sustainability, doi:10.3390/su152416558_

Round 1

Reviewer 1 Report

Comments and Suggestions for Authors

The paper investigates the relationship between urban population size and carbon emissions in the aviation industry, demonstrating a commendable choice of topic. The research is interesting. However, the article exhibits certain shortcomings in both methodology and uncertainty analysis, thus recommending a substantial major revision.

 1. The carbon emissions from aviation journeys are not solely linked to engine output but also depend on factors such as payload and compliance, which the article fails to address.

2. Moreover, there have been notable changes in fuel standards over the past decade. How have these changes affected carbon emission coefficient standards?

3. Additionally, it is imperative for the authors to acknowledge the uncertainties associated with this calculation method. What is the margin of error for these calculations? This would greatly aid readers in assessing the validity of the data.

4. Focusing solely on the total volume of aviation carbon emissions is insufficient. Could the authors consider calculating per-passenger aviation carbon emissions based on passenger traffic? This approach would facilitate the assessment of aviation carbon emission efficiency and the efficiency enhancements brought about by intelligent aviation system management.

5. Addressing the allocation of carbon emissions between cities solely on territorial principles is insufficient. Could a consideration of demand-based principles be explored? Reference to the gravity model from international trade theory, with population size as a weighting factor, could be beneficial in decomposing Eiccd emissions.

Comments on the Quality of English Language

None.

Author Response

1. The carbon emissions from aviation journeys are not solely linked to engine output but also depend on factors such as payload and compliance, which the article fails to address. Response: Thank you for your insightful feedback regarding the consideration of payload and compliance in calculating aviation carbon emissions. I acknowledge the significance of these factors in influencing carbon emissions from aviation journeys. However, when considering payload, we need it's important to note that currently there is no established or universally accepted methodology to accurately quantify the impact of payload and compliance on carbon emissions in aviation. Incorporating such parameters would introduce complexities and potential inaccuracies given the lack of a standardized approach. That being said, I deeply appreciate the emphasis on the comprehensiveness of the research, and will certainly highlight this limitation in the revised manuscript. This mention will also provide future researchers with a direction for further exploration. 2. Moreover, there have been notable changes in fuel standards over the past decade. How have these changes affected carbon emission coefficient standards? Response: Thank you for your insightful feedback,fuel standards indeed will influence carbon emissions from civil aviation. However, to the best of our knowledge, aviation fuel in China has not changed over the past few decades. Due to high costs, high fuel density, and strong externalities, the aviation industry is considered one of the most challenging sectors to transform. Sustainable Aviation Fuel (SAF), electric aircraft, and hydrogen aviation are the three main focal areas for developing green aviation. Among the three, sustainable fuel is the direction relevant to this paper. Currently, airports such as Indira Gandhi International Airport in Delhi, Oslo Airport in Norway, Stockholm Airport in Sweden, Los Angeles Airport in the US, and Seattle Airport have begun supplying SAF. Due to a lack of policy subsidies, China's SAF industry is largely in the knowledge reserve stage. Compared to over 400,000 SAF test flights in the international aviation sector, the number of test flights in China can be counted on one hand. Between 2011 and 2022, the number of test flights did not exceed 10. As a typical policy-driven industry, the path and measures for China to develop SAF are not clear. Compared to the explicit blending directives and sustainable transport fuel application goals in Europe and the US, the legal policies and standard systems issued by the Civil Aviation Administration and other relevant departments are also not well-developed. On July 14, 2023, Air China completed the country's first wide-body domestic sustainable aviation fuel passenger flight. Therefore, we believe that currently, there is no way, nor is it necessary, to consider changes in fuel in this paper . 3. Additionally, it is imperative for the authors to acknowledge the uncertainties associated with this calculation method. What is the margin of error for these calculations? This would greatly aid readers in assessing the validity of the data. Response: Thank you for your insightful feedback. We have added Section 5.3, "Limitations and Future Research Directions", where we present the limitations of our approach. Indeed, the prevailing methodologies predominantly utilize ICAO's bottom-up approach for the LTO phase. Due to data constraints in the CCD phase, most calculations are made based on fuel consumption, leading to relatively crude calculations. Drawing upon the method suggested by Eskenazi (2022), we have undertaken a more precise bottom-up carbon emission calculation for the CCD phase, taking into account various aircraft models and engines. 4. Focusing solely on the total volume of aviation carbon emissions is insufficient. Could the authors consider calculating per-passenger aviation carbon emissions based on passenger traffic? This approach would facilitate the assessment of aviation carbon emission efficiency and the efficiency enhancements brought about by intelligent aviation system management. Response: Thank you for your insightful feedback,this question is something like the first question. Certainly, it would be better to calculate the carbon emissions per passenger for each flight, and the ICAO (Version 10 - June 2017) provides the formula to calculate the average carbon dioxide emissions per passenger. However, due to data limitations in China, we cannot obtain the number of passengers for each flight, making it impossible to compute emissions on a per-passenger basis. To the best of our knowledge, current research has not yet achieved studies based on emissions by passenger count for China. In the further study, if data available, it will be a great idea to calculate per-passenger aviation carbon emissions. 5. Addressing the allocation of carbon emissions between cities solely on territorial principles is insufficient. Could a consideration of demand-based principles be explored? Reference to the gravity model from international trade theory, with population size as a weighting factor, could be beneficial in decomposing Eiccd emissions. Response: Thank you for your insightful feedback. If the intention is to analyze the causative factors behind carbon emissions in different cities from civil aviation, employing the gravity model would indeed be apt. However, the primary focus of this paper is the quantification of city carbon emissions, and a direct equal distribution between cities seems more appropriate. In fact, the prevailing approach for allocating carbon emissions from flights to cities is to distribute them evenly between the two cities in question (Li et al., 2022).

Reviewer 2 Report

Comments and Suggestions for Authors

Dear Author(s),

Article is well structured and the topic is interesting. However, following comments should be addressed prior to further processing of the article.

1)      Refer to abstract: What is CCD and LTO here? Is CCD for Cruise, Climb and Descent LTO for Landing and Take-off? Ensure that each short form is described at its first occurrence.

2)      Refer to abstract: Is it CO2 or CO2?

3)      Refer to abstract, line # 20: It is not good to use “They” here. Recheck and update the sentence.

4)      Refer to abstract, line # 22: Word “bolstering” seems to be generated using AI. If so, then authors must include acknowledgement in the article.  

5)      Refer to abstract, lien # 23: This sentence also seems to be generated using AI “Collectively, this research paints a holistic picture of the complexities and challenges tied to aviation carbon emissions amid China's urban expansion.”

6)      Refer to whole article: Carbon emission is a global issue however, scope of this study is very limited i.e. civil aviation of China’s megacities. How would authors justify it?

7)      Refer to line # 28: What is written at this line? It looks that article has not been properly reviewed by authors prior to submission. Careful review of the whole article is required.

8)      Refer to whole article: Check space between reference number and text. Space is missing everywhere like transport[1], increased by 66%[2,3]., China in 2022[4].

9)      Refer to line # 145: Authors have focused on calculations for 2012-2021. Why not upto 2023? Study seems old.

10)  Refer to line # 179: Here again “From 2013 to 2021”. Why noy upto 2023?

11)  Refer to line 187: Check “respectively, respectively, compared” here.   

12)  Refer to figure 1: What is data source?

13)  Refer to sub-section 2.4: Authors are encouraged to include a comparison table for model-oriented carbon emission of aircrafts.  

14)  Refer to line # 281 and 282: Authors have mentioned that “The initial cycle, LTO, encapsulates all flight activities occurring within an altitude of 3,000 feet (equivalent to 915 meters) from the ground. In contrast, the CCD represents all activities that transpire beyond this 3,000-foot altitude.” What is difference between 3000 feet and 3000-foot? Recheck the altitude values and update accordingly otherwise justify your statement.

15)  Refer to all equations: Check the alignment of all equations.

16)  Refer to line # 397: Why it is highlighted?

17)  Refer to results: Are the results merely calculations? If so, then it is a simple calculation task. Where is the novelty in this study?

18)  Refer to figures 7, 8 and 9: Emission of 5 pollutants is shown in figures 7 and 8 whereas figure 9 shows 4 emissions. Ensure that results are consistent throughout study.   

19)  Refer to lines # 484 – 519: Very long paragraph.

Good luck.    

Comments on the Quality of English Language

Dear Author(s),

Article is well structured and the topic is interesting. However, following comments should be addressed prior to further processing of the article.

1)      Refer to abstract: What is CCD and LTO here? Is CCD for Cruise, Climb and Descent LTO for Landing and Take-off? Ensure that each short form is described at its first occurrence.

2)      Refer to abstract: Is it CO2 or CO2?

3)      Refer to abstract, line # 20: It is not good to use “They” here. Recheck and update the sentence.

4)      Refer to abstract, line # 22: Word “bolstering” seems to be generated using AI. If so, then authors must include acknowledgement in the article.  

5)      Refer to abstract, lien # 23: This sentence also seems to be generated using AI “Collectively, this research paints a holistic picture of the complexities and challenges tied to aviation carbon emissions amid China's urban expansion.”

6)      Refer to whole article: Carbon emission is a global issue however, scope of this study is very limited i.e. civil aviation of China’s megacities. How would authors justify it?

7)      Refer to line # 28: What is written at this line? It looks that article has not been properly reviewed by authors prior to submission. Careful review of the whole article is required.

8)      Refer to whole article: Check space between reference number and text. Space is missing everywhere like transport[1], increased by 66%[2,3]., China in 2022[4].

9)      Refer to line # 145: Authors have focused on calculations for 2012-2021. Why not upto 2023? Study seems old.

10)  Refer to line # 179: Here again “From 2013 to 2021”. Why noy upto 2023?

11)  Refer to line 187: Check “respectively, respectively, compared” here.   

12)  Refer to figure 1: What is data source?

13)  Refer to sub-section 2.4: Authors are encouraged to include a comparison table for model-oriented carbon emission of aircrafts.  

14)  Refer to line # 281 and 282: Authors have mentioned that “The initial cycle, LTO, encapsulates all flight activities occurring within an altitude of 3,000 feet (equivalent to 915 meters) from the ground. In contrast, the CCD represents all activities that transpire beyond this 3,000-foot altitude.” What is difference between 3000 feet and 3000-foot? Recheck the altitude values and update accordingly otherwise justify your statement.

15)  Refer to all equations: Check the alignment of all equations.

16)  Refer to line # 397: Why it is highlighted?

17)  Refer to results: Are the results merely calculations? If so, then it is a simple calculation task. Where is the novelty in this study?

18)  Refer to figures 7, 8 and 9: Emission of 5 pollutants is shown in figures 7 and 8 whereas figure 9 shows 4 emissions. Ensure that results are consistent throughout study.   

19)  Refer to lines # 484 – 519: Very long paragraph.

Good luck.    

Author Response

Article is well structured and the topic is interesting. However, following comments should be addressed prior to further processing of the article.

1)      Refer to abstract: What is CCD and LTO here? Is CCD for Cruise, Climb and Descent LTO for Landing and Take-off? Ensure that each short form is described at its first occurrence.

Response:

Thank you for your suggestion. We have given the full name of CCD and LTO at the abstract and its first occurrence in main text.

2)      Refer to abstract: Is it CO2 or CO2?

Response:

Thank you for your suggestion. We have double checked and revised CO2 in our manuscript.

3)      Refer to abstract, line # 20: It is not good to use “They” here. Recheck and update the sentence.

Response:

Thank you for your suggestion. We have changed the sentence “The implications of this study are profound. They emphasize the urgency for advancements in aviation fuel technology, rigorous management of CCD phase pollutants, strategic carbon emission controls in populous cities” to “The implications of this study emphasize the urgency for advancements in aviation fuel technology, rigorous management of CCD phase pollutants, strategic carbon emission controls in populous cities”.

4)      Refer to abstract, line # 22: Word “bolstering” seems to be generated using AI. If so, then authors must include acknowledgement in the article.  

Response:

Thank you for your suggestion. For some paragraphs, we used GPT to polish. We add an acknowledgement in the end of the main text. “Acknowledgment: The authors use Chat GPT to polish some paragraphs of the main text.”

5)      Refer to abstract, lien # 23: This sentence also seems to be generated using AI “Collectively, this research paints a holistic picture of the complexities and challenges tied to aviation carbon emissions amid China's urban expansion.”

Response:

Thank you for your suggestion. Indeed, we used GPT to polish the language, but the first author wrote every sentence of this manuscript herself, including this sentence. And we have added the acknowledgement already.

6)      Refer to whole article: Carbon emission is a global issue however, scope of this study is very limited i.e. civil aviation of China’s megacities. How would authors justify it?

Response: Thank you for your suggestion. Carbon emission from civil aviation only accounts for a small fraction of the total carbon emission. However, this topic remains of significant importance for the following reasons: (1) Unlike ground emission sources, aircraft exhaust discharges directly into the upper troposphere and stratosphere, causing a more serious greenhouse effect (Lyu et al., 2023). (2) The growth of carbon emissions in civil aviation is rapid. the International Civil Aviation Organization (ICAO) predicts the total greenhouse gases emissions from aviation industry increase by 400–600% in 2050 compared with that in 2010 (Liu et al., 2019).(3)China is the country with the fastest growth in civil aviation carbon emissions and has become the third-largest emitter in the world.(Graver et al., 2020). From 2013 to 2019, China’s carbon emission increased by 66%(Graver et al., 2020; Habib et al., 2021). Among them, the carbon emissions from civil aviation in China's mega-cities account for over 60% of the total civil aviation carbon emissions in China. Therefore, the key to controlling China's civil aviation carbon emissions lies in controlling the emissions from these mega-cities. This is precisely the significance of this article. We have mentioned point 2 and point 3 in our introduction to illustrate the importance of our topic.

Reference:

Baxter, G., Srisaeng, P., & Wild, G. (2020). Airport related emissions and their impact on air quality at a major Japanese airport: The case of Kansai international airport. Transport and Telecommunication, 21(2), 95–109.

Graver, B., Rutherford, D., & others. (2020). CO2 emissions from commercial aviation: 2013, 2018, and 2019.

Habib, Y., Xia, E., Hashmi, S. H., & Ahmed, Z. (2021). The nexus between road transport intensity and road-related CO2 emissions in G20 countries: An advanced panel estimation. Environmental Science and Pollution Research, 28(41), 58405–58425. https://doi.org/10.1007/s11356-021-14731-7

Kurniawan, J. S., & Khardi, S. (2011). Comparison of methodologies estimating emissions of aircraft pollutants, environmental impact assessment around airports. Environmental Impact Assessment Review, 31(3), 240–252. https://doi.org/10.1016/j.eiar.2010.09.001

Li, F., Li, F., Cai, B., & Lv, C. (2022). Mapping carbon emissions of China’s domestic air passenger transport: From individual cities to intercity networks. Science of The Total Environment, 851, 158199. https://doi.org/10.1016/j.scitotenv.2022.158199

Liu, H., Tian, H., Hao, Y., Liu, S., Liu, X., Zhu, C., Wu, Y., Liu, W., Bai, X., & Wu, B. (2019). Atmospheric emission inventory of multiple pollutants from civil aviation in China: Temporal trend, spatial distribution characteristics and emission features analysis. Science of The Total Environment, 648, 871–879. https://doi.org/10.1016/j.scitotenv.2018.07.407

Lyu, C., Liu, X., Wang, Z., Yang, L., Liu, H., Yang, N., Xu, S., Cao, L., Zhang, Z., Pang, L., Zhang, L., & Cai, B. (2023). An emissions inventory using flight information reveals the long-term changes of aviation CO2 emissions in China. Energy, 262, 125513. https://doi.org/10.1016/j.energy.2022.125513

Unal, A., Hu, Y., & Chang, M. E. (2005). Airport related emissions and impacts on air quality: Application to the Atlanta International Airport. Atmospheric Environment, 39(32), 5787-5798.

Zhou, W., Wang, T., Yu, Y., Chen, D., & Zhu, B. (2016). Scenario analysis of CO2 emissions from China’s civil aviation industry through 2030. Applied Energy, 175, 100–108. https://doi.org/10.1016/j.apenergy.2016.05.004

7)      Refer to line # 28: What is written at this line? It looks that article has not been properly reviewed by authors prior to submission. Careful review of the whole article is required.

Response:

Thank you for your careful review, but the line should be added by the editor. In the version we submitted there was no line. We speculate that lines were added due to the formatting requirements of SUS, which happen to be right above the GEL classification.

8)      Refer to whole article: Check space between reference number and text. Space is missing everywhere like transport[1], increased by 66%[2,3]., China in 2022[4].

Response:

Thank you for your careful review, we have revised all the missing space.

9)      Refer to line # 145: Authors have focused on calculations for 2012-2021. Why not up to 2023? Study seems old.

Response: Thank you for your suggestions. Although we also would like to update the date, due to the lag in statistical information, 2021 is already the latest year for which data can be obtained.

10)  Refer to line # 179: Here again “From 2013 to 2021”. Why not up to 2023?

Response: Thank you for your suggestions. Although we also would like to update the date, due to the lag in statistical information, 2021 is already the latest year for which data can be obtained.

11)  Refer to line 187: Check “respectively, respectively, compared” here.  

 Response:

Thank you for your careful review, we have deleted one of “respectively,”.

12)  Refer to figure 1: What is data source?

 Response:

Thank you for your review comments, we have revised the first sentence of this paragraph, as “According to the flight schedule data from the Civil Aviation Administration of China (CAAC), from 2013 to 2021, there is a discernible upward trend in the total number of national civil aviation flights, climbing from 2.95 million to 5.2 million instances, as depicted in Figure 1.” And also added a “Note” to present the data source for Figure 1.

13)  Refer to sub-section 2.4: Authors are encouraged to include a comparison table for model-oriented carbon emission of aircrafts.  

 Response:

     Thank you for your suggestion. In Section 2.4, we provide a tabulated representation elucidating the distribution of various aircraft types. However, it's pertinent to note that Section 2.4 merely delineates the scope of our research and the methodologies employed. The comparative tabulation detailing the model-centric carbon emissions of aircrafts inherently embodies the computational outcomes. Consequently, we have allocated the findings concerning emissions attributable to distinct aircraft models to Section 4.3, titled 'Carbon Emissions from Different Aircraft Types.

14)  Refer to line # 281 and 282: Authors have mentioned that “The initial cycle, LTO, encapsulates all flight activities occurring within an altitude of 3,000 feet (equivalent to 915 meters) from the ground. In contrast, the CCD represents all activities that transpire beyond this 3,000-foot altitude.” What is difference between 3000 feet and 3000-foot? Recheck the altitude values and update accordingly otherwise justify your statement.

Response:

Thank you for your suggestion, we have revised 3000-foot to 3000 feet.

15)  Refer to all equations: Check the alignment of all equations.

 Response:

Thank you for your suggestion, we have double checked and revised the alignment of all equations.

16)  Refer to line # 397: Why it is highlighted?

Response:

Thank you for your suggestion, we have deleted the repeated sentence.

17)  Refer to results: Are the results merely calculations? If so, then it is a simple calculation task. Where is the novelty in this study?

 Response:

Thank you for your suggestion. Indeed, this paper focuses solely on calculating carbon emissions from China's civil aviation, but its significance is manifold. Firstly, the workload was immense. We had to gather data on flight departures between 2012 and 2021. Secondly, it was necessary to collect data on the matching of aircraft types with their engines. Lastly, we had to gather emission factors for different engines during both the LTO and CCD phases to compute the carbon emissions for each flight. The importance of this work lies in: To analyze the future emission scenarios of the aviation industry, explore the optimal emission pathways, implement detailed emission reduction measures, and formulate a carbon peak plan, it is essential for us to accurately calculate the carbon dioxide emissions from aviation and establish an inventory of atmospheric pollutants and carbon emissions for civil aircraft.

18)  Refer to figures 7, 8 and 9: Emission of 5 pollutants is shown in figures 7 and 8 whereas figure 9 shows 4 emissions. Ensure that results are consistent throughout study.   

Response:

Thank you for your suggestion, we have made modifications to Figure 9 and added the fifth type of pollutant.

19)  Refer to lines # 484 – 519: Very long paragraph.

Response:

Thank you for your suggestion, we have divided it into two part.

Reviewer 3 Report

Comments and Suggestions for Authors

The presented article is devoted to a highly topical issue. Many scientists and specialists today are investigating the problem of global warming and greenhouse gas emissions. The authors approach is interesting, however, despite the appreciation of the article, I strongly recommend to improve the following issues::

1.     In my opinion a few more key words should have been added

2.     The introduction should be expanded, or literature review section should be added. The reference list consists of only 23 sources, 6 of which are scientific articles with a publication date earlier than 2019, and 5 sources are not scientific literature but sources of statistical or other information. It have to be explained how cited references (scientific articles) were selected or improved or add the sections mentioned above.

3.     On the one hand, the article fits the scope of the journal, but the direct link between sustainable development and carbon dioxide emissions is not specified, perhaps it would be worthwhile in the introduction to talk about China's goals to reduce carbon dioxide emissions, the attention of society to this problem and then move on to consider the aviation industry as a major emitter. The relationship between economic growth and carbon dioxide emissions should also be highlighted in more detail to fully reflect the theme of the special issue

4.     It is reasonable to formulate and clearly present the purpose of the study, the research gap and raise the research question. It is necessary to present the discussion section. In the discussion section it is necessary to present the authors results obtained, and how they differ from previously existing ones.

5.     The study presents several formulas that the authors use in their calculations. However, the original source of the formulas is not clear, the text refers to modern scientific publications and it is unlikely that their authors are the developers of these formulas

6.     Please check the numbering of figures and their captions. So after figure 2 comes figure 5. The caption of figure 5 (a and b-line 390 and 392) covers the year 2022, but the graph only shows the year 2021

7.     in the implications section, the authors offer targeted guidance for emission reductions However, it is not explained why the authors focused on these directions and other possible options are not presented or analyzed.

Author Response

  1. In my opinion a few more key words should have been added

Response:

Thank you for your suggestion, we have added two more keywords:LTO phase; CCD phase.

  1. The introduction should be expanded, or literature review section should be added. The reference list consists of only 23 sources, 6 of which are scientific articles with a publication date earlier than 2019, and 5 sources are not scientific literature but sources of statistical or other information. It have to be explained how cited references (scientific articles) were selected or improved or add the sections mentioned above.

Response:

Thank you for your suggestion, we have expanded both the introduction and literature review section. The reference list has been expanded to 38 now.

  1. On the one hand, the article fits the scope of the journal, but the direct link between sustainable development and carbon dioxide emissions is not specified, perhaps it would be worthwhile in the introduction to talk about China's goals to reduce carbon dioxide emissions, the attention of society to this problem and then move on to consider the aviation industry as a major emitter. The relationship between economic growth and carbon dioxide emissions should also be highlighted in more detail to fully reflect the theme of the special issue

Response:

Thank you for your suggestion, in the first paragraph of the introduction, we have incorporated two recent aviation carbon reduction plans from China to present the Chinese government's objectives for aviation carbon reduction, and to illustrate the connection between sustainable development and carbon dioxide emissions. We've also added content to show the connection between sustainable development and carbon dioxide emissions. In addition, we highlighted the impact of China's new urbanization development on aviation carbon emissions as follows:

“As urban centers expand and more cities emerge as key hubs, air connectivity between these cities has grown, driven by both business and leisure travel demands. New airports and routes have been introduced to accommodate this surge in demand. As a direct consequence, flight frequencies have increased by approximately 10% annually over the past decade, leading to elevated levels of aviation carbon emissions. While new airports and routes accommodate this growth, their construction and energy-intensive processes also contribute to emissions. According to the “14th Five-Year Plan for Civil Aviation Development”, the Compound Annual Growth Rate (CAGR) of China's civil aviation passenger traffic from 2019 to 2025 is projected to be 5.9%. By 2025, the passenger traffic is expected to reach 930 million [12]. Currently, China's civil aviation industry is still in a growth stage, and with the further urbanization, there is significant potential for increasing the per capita air travel frequency. Thus, an interesting question has been posed: What are the spatiotemporal trends in carbon emissions in China's civil aviation sector? Specifically, as China's rapidly developing mega-cities have a substantial de-mand for civil aviation, how fast are carbon emissions from aviation growing in these metropolises?"

  1. It is reasonable to formulate and clearly present the purpose of the study, the research gap and raise the research question. It is necessary to present the discussion section. In the discussion section it is necessary to present the authors results obtained, and how they differ from previously existing ones.

Response:

Thank you for your suggestion. We have introduced and refined our research question in the opening paragraph of the introduction: “Thus, an intriguing question emerges: What are the spatiotemporal trends in carbon emissions within China's civil aviation sector? Specifically, with China's rapidly growing megacities exhibiting substantial demand for civil aviation, how has the growth rate of carbon emissions from these megacities evolved?”

  1. The study presents several formulas that the authors use in their calculations. However, the original source of the formulas is not clear, the text refers to modern scientific publications and it is unlikely that their authors are the developers of these formulas.

   Response:

Thank you for your suggestion. In the literature review, we clarified the two commonly used methods for calculating civil aviation carbon emissions, along with the literature that employs these methods. In fact, most papers have adopted the top-down approach of ICAO or the bottom-up method by EEA based on fuel consumption, or improvements upon these foundational methods.“The advanced approach of the ICAO, considering the pollutant emissions from different aircraft and engine types at various flight stages, is the most widely used, deemed the "bottom-up" method. This methodology is similar to that of the U.S. EPA(Baxter et al., 2020; Unal et al., 2005; Zhou et al., 2016). However, the ICAO method focuses only on the emissions during the Landing and Takeoff (LTO) phase. According to the International Air Transport Association (IATA) data, LTO cycles account for about 10% of all emissions during flight(Kurniawan & Khardi, 2011). Another method widely used is the European Monitoring and Evaluation Program (EMEP) by the European Environment Agency (EEA). This method, considered a "top-down" approach, estimates CO2 emissions based on total fuel consumption during flight, without the need to consider differences between different flight engines. Its advantage lies in its convenience, as it doesn't require detailed information about flight duration, aircraft type, or engine. However, its precision is somewhat lacking. ” This study follows the ICAO method for the LTO phase, and utilizes Eskenazi (2022)'s approach to calculate emissions during the CCD phase.

  1. Please check the numbering of figures and their captions. So after figure 2 comes figure 5. The caption of figure 5 (a and b-line 390 and 392) covers the year 2022, but the graph only shows the year 2021

Response:

Thank you for your suggestion, we have double checked the figures and their captions, and fixed the problems.

  1. in the implications section, the authors offer targeted guidance for emission reductions. However, it is not explained why the authors focused on these directions and other possible options are not presented or analyzed.

Response:

Thank you for your suggestion. In regards to the implications section, our intent was to highlight specific guidance for emission reductions based on the most pressing findings from our research. However, we appreciate your insight into the importance of explaining the rationale behind our focus and the potential value of discussing other alternatives. In light of your feedback, we plan to revise the section to add some research outlook. This, we believe, will provide a more comprehensive understanding to the readers and enrich the overall discussion.

Round 2

Reviewer 1 Report

Comments and Suggestions for Authors

All the problem mentioned are well addressed. 

Comments on the Quality of English Language

None

Author Response

Thank you for your review comments, which have helped improve the quality of our article.

Reviewer 2 Report

Comments and Suggestions for Authors

Dear Author(s),

Good effort. Most of my comments are satisfactorily addressed however following comments need author’s attention.

1)      Refer to line # 28: What is written at this line?

2)      Refer to figure 3, 4, 5, 7: Check CO2 here. You should recheck the whole article again.

Good luck.    

Comments on the Quality of English Language

Dear Author(s),

Good effort. Most of my comments are satisfactorily addressed however following comments need author’s attention.

1)      Refer to line # 28: What is written at this line?

2)      Refer to figure 3, 4, 5, 7: Check CO2 here. You should recheck the whole article again.

Good luck.    

Author Response

Thank you for your kind review and constructive comments regarding our manuscript. We have taken careful note of the additional feedback you have provided and have made the requisite amendments as outlined below:

  1. As per your query regarding line #28: The text mentioned corresponds to the JEL (Journal of Economic Literature) code. In alignment with the standard formatting adopted by the SUS, we have now positioned the JEL code above the horizontal line, as indicated.
  2. Concerning figures 3, 4, 5, and 7, we have revised the representation of CO2 and re-examined the entire manuscript for consistency and accuracy. Any modifications that have been made subsequent to your feedback have been highlighted in red for ease of identification and review.

Reviewer 3 Report

Comments and Suggestions for Authors

Thanl you for improvements

Author Response

(The authors gave the same response as above.)
